# In the Beginning: Let Hydration Be Coded in Proteins for Manifestation and Modulation by Salts and Adenosine Triphosphate

**DOI:** 10.3390/ijms252312817

**Published:** 2024-11-28

**Authors:** Jianxing Song

**Affiliations:** Department of Biological Sciences, Faculty of Science, National University of Singapore, 10 Kent Ridge Crescent, Singapore 119260, Singapore; dbssjx@nus.edu.sg or songjianxing1000@gmail.com; Tel.: +65-65161013; Fax: +65-67792486

**Keywords:** water, ATP, protein hydration, protein folding, intrinsically disordered regions (IDRs), protein phase separation, cellular networks, artificial intelligence (AI)

## Abstract

Water exists in the beginning and hydrates all matter. Life emerged in water, requiring three essential components in compartmentalized spaces: (1) universal energy sources driving biochemical reactions and processes, (2) molecules that store, encode, and transmit information, and (3) functional players carrying out biological activities and structural organization. Phosphorus has been selected to create adenosine triphosphate (ATP) as the universal energy currency, nucleic acids for genetic information storage and transmission, and phospholipids for cellular compartmentalization. Meanwhile, proteins composed of 20 α-amino acids have evolved into extremely diverse three-dimensional forms, including folded domains, intrinsically disordered regions (IDRs), and membrane-bound forms, to fulfill functional and structural roles. This review examines several unique findings: (1) insoluble proteins, including membrane proteins, can become solubilized in unsalted water, while folded cytosolic proteins can acquire membrane-inserting capacity; (2) Hofmeister salts affect protein stability by targeting hydration; (3) ATP biphasically modulates liquid–liquid phase separation (LLPS) of IDRs; (4) ATP antagonizes crowding-induced protein destabilization; and (5) ATP and triphosphates have the highest efficiency in inducing protein folding. These findings imply the following: (1) hydration might be encoded in protein sequences, central to manifestation and modulation of protein structures, dynamics, and functionalities; (2) phosphate anions have a unique capacity in enhancing μs-ms protein dynamics, likely through ionic state exchanges in the hydration shell, underpinning ATP, polyphosphate, and nucleic acids as molecular chaperones for protein folding; and (3) ATP, by linking triphosphate with adenosine, has acquired the capacity to spacetime-specifically release energy and modulate protein hydration, thus possessing myriad energy-dependent and -independent functions. In light of the success of AlphaFolds in accurately predicting protein structures by neural networks that store information as distributed patterns across nodes, a fundamental question arises: Could cellular networks also handle information similarly but with more intricate coding, diverse topological architectures, and spacetime-specific ATP energy supply in membrane-compartmentalized aqueous environments?

## 1. Introduction

The book of Genesis in the Bible begins with the verse: “Now the earth was formless and empty, darkness was over the surface of the deep, and the Spirit of God was hovering over the waters” (Genesis 1:2). Water (H_2_O) is thought to have formed in the early universe when hydrogen, the simplest and most abundant element, combined with oxygen produced in the interiors of stars, creating water molecules in interstellar space. Although water is abundant in the cosmos, not all planets exist under the right conditions to retain or accumulate it. A planet’s ability to host water depends on factors such as its formation process, proximity to its star, size, and geological history [1]. Covering two-thirds of our planet, water is a unique and ubiquitous substance in our world and has by all accounts mysterious properties that set it aside from other molecules [2]. At least to our experience, life is invariably linked to the presence of water, as the earliest life forms likely originated in aqueous environments [2,3,4]. Often referred to as the ‘matrix of life’, water is not merely a passive backdrop but plays active and crucial roles in molecular biology [4].

Hydration is a fundamental and universal phenomenon that is common to all classes of compounds regardless of organic and inorganic chemicals [5,6,7,8,9]. It refers to the interaction between water molecules and other substances, which can occur through various mechanisms such as hydrogen bonding, electrostatic interactions, and van der Waals forces. The extent and nature of hydration depend on the chemical properties of the substances involved, such as polarity, charge, and molecular structure. Inorganic chemicals, particularly salts, ions, and minerals, often exhibit hydration through the formation of hydration shells around ions [5,6]. Organic molecules can also undergo hydration, particularly those with functional groups capable of hydrogen bonding or those that are polar. Hydration can significantly influence the solubility, stability, and reactivity of organic molecules in aqueous environments [7,8,9]. It is well-established that water mediates or even governs numerous vital biological processes such as protein folding, enzyme catalysis, membrane self-assembly, and substrate recognition through the hydration of a wide range of biomacromolecules and small molecules, including salts.

Proteins are the most essential functional players in almost all biological processes within living cells and even viruses [10]. Their multifaceted roles highlight their indispensability for life, encompassing structural support, intricate biochemical regulation, and defense mechanisms. Key functions include the following: (1) Structural Support: collagen, the most abundant protein in the human body, constitutes ~30% of total protein content and forms the extracellular matrix in connective tissues such as skin, tendons, and bones. (2) Enzymatic Activity: enzymes accelerate biochemical reactions by lowering activation energy, enabling reactions to occur millions of times faster. (3) Transport and Storage: proteins facilitate the movement of substances across membranes or within the body. For example, hemoglobin in red blood cells binds oxygen, with ~98.5% of blood oxygen transported via hemoglobin. (4) Immune Defense: proteins are vital to immunity. Antibodies (immunoglobulins), produced by B-cells, exist in billions of unique forms adapted to specific antigens. Complement proteins (~30 types) enhance pathogen destruction via opsonization, lysis, or inflammation. (5) Signaling: proteins function as hormones or receptors, enabling intercellular communication. Insulin, a 51-amino-acid protein hormone, regulates blood glucose levels. Receptors like G-protein-coupled receptors (over 800 types in humans) detect signals and initiate intracellular responses. (6) Movement: motor proteins, such as myosin and actin, drive cellular and organismal movement. In muscles, myosin heads interact with actin filaments, generating contraction forces at a rate of 5–6 ATP molecules per second. (7) Regulation: proteins modulate gene expression and metabolic pathways. For example, transcription factor p53, a tumor suppressor, regulates genes in response to stress, influencing apoptosis, the cell cycle, and DNA repair. (8) Energy Source: proteins yield ~4 kcal/g of energy when metabolized, particularly during fasting or intense exercise. (9) Buffers: proteins like hemoglobin help maintain pH homeostasis, binding hydrogen ions to stabilize blood pH around 7.4 [10].

As illustrated in Figure 1A, proteins are composed of 20 α-amino acids, each with distinct physicochemical properties. Based on their amino acid sequences, proteins encoded by higher eukaryotic genomes can be categorized into two main types [11]: those with random, high-complexity sequences (I of Figure 1A) and those with non-random, low-complexity sequences (II of Figure 1A). Interestingly, in cell-like buffers containing high salt concentrations, only a subset of proteins within the first category can fold into well-defined, soluble structures [12,13,14,15]. Conversely, many proteins in the second category remain fully functional despite lacking well-defined structures, and are thus referred to as intrinsically disordered proteins (IDPs) [15,16,17]. Additionally, approximately 25–30% of proteins are associated with phospholipids [18], earning them the designation of membrane proteins (MPs) (Figure 1A).

Water has been proposed to “slave” proteins, implying that the dynamic behavior of proteins is intricately coupled with the dynamics of their surrounding water molecules, specifically the hydration shell [20,21,22,23,24]. Protein hydration involves the interaction of water molecules with the protein’s surface. These water molecules form hydrogen bonds with polar and charged groups, engage in hydrophobic interactions, and significantly influence protein folding, stability, and dynamics. Through hydrogen bonding, hydrophobic interactions, electrostatic screening, and dynamic exchange, water plays a crucial role in controlling protein behavior, facilitating folding, conformational changes, and allosteric regulation. This intricate interplay underscores the essential role of water in the life and functionality of proteins [25,26,27,28,29].

Nevertheless, despite exhaustive studies, it still remains extremely challenging to understand protein hydration using both experimental and computational methods due to the complexity and dynamic nature of the interactions between water molecules and protein surfaces. These complexities include the following: (1) the dynamic and heterogeneous nature of hydration, which involves a rapidly changing network of water molecules that can form and break hydrogen bonds in picoseconds. In this regard, techniques such as nuclear magnetic resonance (NMR) spectroscopy and X-ray crystallography provide only static or averaged views, making it difficult to capture the full dynamic picture [20]. On the other hand, simulating protein hydration requires capturing fast and localized interactions between water and protein, necessitating a high temporal and spatial resolution. Molecular dynamics (MD) simulations must balance accuracy with computational feasibility [30]. (2) Complex interactions and multiplicity of states. Therefore, different techniques often provide complementary but not complete information. For instance, X-ray crystallography can pinpoint positions of water molecules tightly bound to proteins in crystal structures, but misses the dynamically interacting water molecules in solution [20]. For simulations, its accurate modeling requires considering many-body interactions, the polarizability of water molecules, and long-range electrostatic effects, which are computationally demanding [20]. (3) Influence of environment and conditions. As protein hydration is sensitive to environmental conditions such as temperature, pH, and ionic strength, reproducing physiological conditions accurately in experiments can be challenging [30,31]. Simulating the exact conditions found in vivo, including the presence of cosolutes and crowding effects, adds layers of complexity to computational models [32]. (4) Scale and resolution limitations. Techniques like neutron scattering and infrared spectroscopy provide insights into hydration but often lack the resolution to distinguish between water molecules bound to specific protein sites [31,33]. MD simulations require balancing fine-grained details (e.g., quantum effects in hydrogen bonding) with the need to simulate large systems over biologically relevant timescales [30].

This review examines several unique findings on protein structures, membrane-interactions, dynamics, and liquid–liquid phase separation (LLPS) and aggregation, which appear to be fundamentally related to protein hydration and its modulation by salt ions and ATP: (1) insoluble proteins, including membrane proteins, can be solubilized in unsalted water, while folded cytosolic proteins can be “unlocked” to release aggregation-prone and membrane-inserting regions; (2) kosmotropic and chaotropic salts affect protein stability and dynamics critically by altering hydration; (3) ATP biphasically modulates LLPS of Arg-/Lys-containing intrinsically disordered regions (IDRs); (4) ATP antagonizes crowding-induced protein destabilization likely by mediating protein hydration; (5) ATP and triphosphates have the highest efficiency in inducing protein folding. The analysis suggests the following: (1) hydration appears to be coded in protein sequences, being central to manifestation and modulation of protein structures, dynamics, and functionalities; (2) phosphate anions have a unique capacity in enhancing μs-ms dynamics of proteins, likely through ionic state exchanges in the hydration shell, underpinning ATP, polyphosphate, and nucleic acids as molecular chaperones for protein folding; and (3) ATP, by linking triphosphate with adenosine, has acquired the capacity to spacetime-specifically release energy and modulate protein hydration, thus possessing myriad energy-dependent and -independent functions.

Finally, three key challenges are discussed in this review. First, molecular mechanisms by which hydration is coded into protein sequences and how protein hydration is modulated remain to be elucidated. Second, in light of the central role of the phosphate-containing molecules in cellular processes, understanding how phosphate ions affect their structures, dynamics, and interactions is essential. In particular, investigations are needed to address whether phosphate anions may govern the dynamics of nucleic acids and phospholipids, which may thus possess currently unknown functions. Third, when considering the success of AlphaFold, a neural-network-based AI system that accurately predicts protein structures by encoding information as distributed patterns across its network nodes, a fundamental question arises: Could cellular networks, which are fundamental to life, also manage information in a similar distributed manner? In particular, cellular networks are far more complex, involving intricate coding mechanisms and diverse topological architectures, and are driven by spacetime-specific ATP energy supplies in membrane-compartmentalized aqueous environments. In this context, could quantum principles be at play, thus leading to the emergence of phenomena such as consciousness.

## 2. Unusual Salt Effect on Protein Solubility Beyond “Salting-In” and “Salting-Out”

Protein solubility is essential not only for researchers and industries that work with proteins in solution, such as structural biologists and the pharmaceutical sector, but also because it is intimately linked to protein aggregation and amyloidogenesis, which are universal hallmarks of an increasing spectrum of human diseases and aging [34,35,36,37,38,39,40,41]. These diseases include neurodegenerative disorders like Alzheimer’s disease (AD), Parkinson’s disease (PD), Huntington’s disease (HD), spinocerebellar ataxias (SCA), amyotrophic lateral sclerosis (ALS), and frontotemporal dementia (FTD) [37,38,39], and even extend to the aging process [40], affecting organisms as diverse as humans and *E. coli* cells [41]. Thermodynamically, solubility is defined as the concentration of protein in a saturated solution that is in equilibrium with its solid phase, whether crystalline or amorphous, under specific conditions. Protein solubility is influenced by both extrinsic and intrinsic factors. Extrinsic factors include pH, ionic strength, temperature, and the presence of solvent additives, which can sometimes be adjusted to improve solubility. However, intrinsic factors, primarily the structures and physicochemical properties on the protein’s surface, also play a significant role in determining solubility [42,43,44,45].

Among the earliest recognized extrinsic factors affecting protein solubility are salts, which are ionic compounds composed of positively charged cations and negatively charged anions, held together by electrostatic forces. When salts dissolve in water, they dissociate into their constituent ions, separating into positively charged cations and negatively charged anions. Salts are quite prevalent and important among the chemicals on Earth due to their various roles in natural processes, industry, and daily life. Many salts naturally exist as minerals, and seawater is especially abundant in salts, with sodium chloride (NaCl) being the most common. Seawater contains approximately 3.5% dissolved salts by weight [46]. In many natural environments, inorganic salts are more prevalent than organic ones. Oceans, soils, and rocks are predominantly composed of inorganic salts. Although organic salts are less common, they are vital in biological systems, functioning as electrolytes and aiding in osmoregulation within the body [47,48].

In 1988, Franz Hofmeister published a landmark study on the influence of salts on protein solubility, laying the groundwork for the concepts of the “salting-in” and “salting-out” effects, as well as the Hofmeister series [49]. In this pioneering study, Hofmeister observed a universal phenomenon: the solubility of proteins in aqueous solutions could be dramatically influenced by salts in a dual manner. At low concentrations (usually <300–500 mM), salts enhance protein solubility (salting-in effect), but reduce it at higher concentrations (salting-out effect), as illustrated in Figure 1B. Hofmeister’s research established the first phenomenological theory for the effects of salts on protein solubility, leading to a dogmatic concept in biochemistry that guides research and practical applications in protein chemistry [50,51,52,53,54].

Intriguingly, a notable subset of proteins appears to be completely insoluble under various buffer conditions, as evidenced by structural genomics projects that cloned and expressed proteins on a proteome-wide scale, but found that nearly half of them were insoluble [55]. This issue is further reflected in eukaryotic cells, which struggle with protein aggregation even under normal, unstressed conditions. Remarkably, studies on human cell lines have estimated that approximately 30% of newly synthesized proteins aggregate and are swiftly degraded by proteasomes [56,57,58,59,60,61]. These proteins likely fail to achieve their native structures due to errors in translation or in post-translational modifications critical for proper folding [56,57,58,59]. However, the intrinsic factors underlying this complete insolubility remained unknown previously.

In 2005, I was inspired to consider that these completely insoluble proteins could actually be solubilized in unsalted water [61]. This idea was later confirmed by the study on 11 proteins that were previously deemed unrefoldable and insoluble in various buffers. Marvelously, all of them were successfully solubilized in unsalted water, allowing high-resolution NMR investigations [62]. The upper limit of this discovery was further explored by examining a 25-residue integral membrane peptide from the influenza M2 channel, one of the most hydrophobic protein sequences known in nature. To the astonishment, this peptide could also be solubilized in unsalted water, forming a highly helical conformation even in the absence of any lipid molecules [63]. Since this discovery, “completely insoluble” mutants and proteins have been characterized, which are implicated in human diseases such as cancers and ALS [63,64,65,66,67,68,69,70,71,72,73]. Later, other research groups have also shown that proteins that were previously considered to be insoluble could, in fact, be solubilized in unsalted water [74,75,76]. Notably, all proteins in the total cellular extract of human cells, including the 30% that are membrane proteins, have been shown to be highly soluble in pure water when the mixture is free of nucleic acids [76]. These results together logically suggest that despite existing in the background, salts play a decisive role in controlling protein aggregation [63].

This discovery allowed us to characterize the high-resolution conformations of these ‘insoluble proteins’ by NMR spectroscopy, which led to the classification of them into four distinct groups (Figure 1C): Group 1 includes proteins that are highly disordered and lack both secondary and tertiary structures [62,64,69,70,71,73,74]; Group 2 consists of proteins that have some helical secondary structures but no tertiary packing [62,65,75]; Group 3 comprises proteins that possess secondary structures to some degrees along with loose tertiary packing, resembling molten globule states [62,63,77]; and Group 4 involves proteins that coexist between the folded and unfolded states, undergoing dynamic exchanges on a millisecond timescale [67,68,72]. Remarkably, all these insoluble proteins have been shown to lack tight tertiary packing.

The results thus revealed a previously unrecognized regime in protein behaviors, where the classic “salting-in/salting-out” principle appears to only apply to well-folded proteins. In contrast, unrefoldable and insoluble proteins, including the most hydrophobic integral membrane protein fragments, can only be solubilized in water with minimal salt concentrations. In 2008, I proposed a model (Figure 1D) to rationalize why these unrefoldable and insoluble proteins remain soluble in unsalted water but aggregate upon the introduction of even small amounts of salt ions [61,62,63]. Briefly, the lack of a tight tertiary structure of these proteins exposes a considerable number of hydrophobic side chains to the surrounding water. When dissolved in water with the minimal salt ions and a pH differing from their isoelectric point (pI), these proteins acquire a significant number of net charges, thus leading to strong electrostatic repulsion between molecules and a substantial hydration shell, which form an energy barrier to inhibit intermolecular interactions. As a result, protein aggregation is largely suppressed in the absence of salts (I in Figure 1D). However, the addition of even small amounts of salt ions can shield these repulsive electrostatic forces and partially disrupt the hydration shell (II in Figure 1D), thus allowing hydrophobic interactions to predominate, which result in pronounced protein aggregation (III in Figure 1D), eventually manifesting as the salting-out phase (IV in Figure 1D).

This model suggests that DRiPs [56,57,58,59], which are “intrinsically insoluble” in vitro, may also inevitably aggregate in vivo due to their inability to fold into stable tertiary structures and the high ion concentrations in cellular environments. Unlike “misfolded proteins”, the aggregation of these “intrinsically insoluble proteins” cannot be prevented by chaperone systems. Therefore, despite the metabolic cost, cells likely degrade these proteins immediately after synthesis to minimize potential damage. If not effectively degraded, their aggregation could trigger various human diseases, including cancers and neurodegenerative disorders, and even contribute to aging.

Protein hydration appears to play a crucial role in solubilization of these insoluble proteins in unsalted water, as well as aggregation or amyloid formation upon the introduction of salts. In unsalted water, the disordered structures of these proteins expose their polypeptide chains extensively, leading to significant hydration. The interaction with a large number of water molecules creates a high energy barrier that prevents the transition to aggregation [78,79,80,81,82]. However, when salt ions are introduced, they neutralize the repulsive forces and disrupt the hydration shell, driving the proteins to form aggregates with a reduced water content [82,83,84]. In this process, the release of ordered water molecules from their association with the proteins into the bulk solvent increases the entropy of the water, which in turn entropically promotes aggregation.

This discovery offers a potential solution to the longstanding ‘chicken-and-egg paradox’ regarding the origin of integral membrane proteins. Specifically, it has been puzzling how these highly hydrophobic, water-insoluble proteins could have reached the early membranes, as “even if occasionally synthesized, they would likely remain stuck in the ribosome” [85]. However, the model suggests a resolution: emerging evidence indicates that proteins and primitive membranes containing integral membrane proteins may have emerged in unsalted, slightly acidic prebiotic oceans [85,86]. Remarkably, these conditions closely resemble the aqueous conditions used to solubilize insoluble or membrane proteins, namely largely unsalted and mildly acidic [87,88,89,90,91,92,93,94,95,96]. Consequently, as illustrated in I of Figure 2A, even the most hydrophobic integral membrane peptides would not be trapped in the early protein-synthesizing machinery. Instead, they would be soluble in this prebiotic, unsalted medium. Furthermore, the concentrations of most proteins in primitive oceans were likely very low, allowing them to diffuse freely and encounter primeval lipid molecules. However, the increase in salt concentrations would drive the spontaneous assembly of lipid molecules and hydrophobic peptides, forming primitive membranes associated with different membrane proteins including integral membrane proteins (II of Figure 2A).

This discovery may also offer insights into another longstanding mystery related to protein diversification. It has been suggested that the space of realized protein folds represents only a fraction, about one-tenth of the possible protein fold space [97], indicating that much of the sequence space remains unexplored in Earth’s life forms. This limited exploration might be linked to ocean salinity. The primitive machinery responsible for protein generation is believed to have existed before the development of membrane-enclosed cells [98]. In unsalted oceans, proteins with the highly randomized sequences could be synthesized in the primitive machinery, all of which are soluble and could freely diffuse. Because in unsalted water, protein–protein interactions are greatly suppressed due to strong electrostatic repulsion between individual protein molecules, a slight increase in ocean salinity might have reduced these repulsive forces, thus facilitating the formation of various protein-based complexes. However, once membrane-enclosed cells emerged and ocean salinity increased, highly hydrophobic proteins would have been prone to aggregation and could potentially damage membranes. This may have driven the evolution of mechanisms that restricted the random sampling of protein sequence space. In this context, modern proteins, regardless of being well-folded, intrinsically disordered, or membrane-associated, likely evolved from a limited pool of primordial proteins that were randomly generated in unsalted oceans. This hypothesis aligns with the proposal that modern proteins appear designed so that their intrinsic repulsive interactions in pure water are sufficient to counteract attractive forces, thereby preventing severe precipitation or aggregation [62,63]. Additionally, it is plausible that the 20 natural L-, α-amino acids were selected because their polymerized forms, proteins, were soluble in the primitive unsalted oceans [63].

## 3. Transformation of the Folded Cytosolic Proteins into Membrane-Interacting Proteins

In cells, where aqueous solutions and membrane systems coexist, proteins are classified as membrane or non-membrane proteins. However, it remains unclear whether non-membrane proteins can transform into membrane proteins. Interestingly, on the other hand, many intrinsically disordered and partially soluble proteins associated with human diseases have been shown to contain regions that can insert into membranes by forming amphiphilic or hydrophobic helices. They include prion protein involved in spongiform transmissible encephalopathies [99], amyloid beta-peptides in Alzheimer’s disease [100], tau tangles in Alzheimer’s disease [101], α-synuclein in Parkinson’s disease [102,103], huntingtin in Huntington’s disease [104], and islet amyloid polypeptide in type II diabetes [105]. Previously, a hydrophobic region within the prion-like domain of TDP-43 was identified as capable of inserting into membranes, transitioning from a partially folded helical conformation to a well-defined Ω–loop–helix motif in vitro [69]. Recently, this region has been shown indeed to be associated with mitochondria [106]. Intriguingly, on the other hand, many well-folded non-membrane proteins have been found to convert into unfolded and insoluble forms by genetic mutations or environmental factors, many of which suddenly become cytotoxic and associated with neurodegenerative diseases and aging. However, the mechanisms underlying their gain of cytotoxicity remained poorly understood.

Previous studies on the unrefoldable and insoluble forms of ALS-related proteins, including P56S-MSP, L126Z-SOD1, nascent SOD1, and C71G-Profilin1 [107], revealed that these insoluble forms are either unfolded or coexist with their unfolded states. Most unexpectedly, these unfolded states have been decoded to gain a novel ability to interact with membranes, driven by the formation of helices or loops over amphiphilic or hydrophobic regions. These regions, which universally exist in all folded proteins, are typically locked in their folded native states. However, upon destabilization or disruption of their folded structures by genetic mutations and environmental damage, these regions become exposed completely or dynamically, thus becoming able to interact with membranes. As a consequence, the mutated or damaged proteins gain cytotoxicity by interacting with membranes, with a mechanism that has no fundamental difference from those underlying the membrane-interacting capacities of amyloid beta-peptides, α-synuclein, and even the membrane toxin melittin. These findings suggest that most, if not all, proteins contain segments capable of folding into distinct structures depending on whether they are in aqueous or membrane environments. Abnormal membrane interactions may initiate disease and aging processes, and when coupled with protein aggregation, can lead to severe proteotoxicity. This may result in the formation of inclusions composed of damaged membranous organelles and protein aggregates, further exacerbating cellular damage.

Indeed, ALS-causing SOD1 mutants were extensively shown to associate tightly with the mitochondrial membrane, behaving similarly to integral membrane proteins, with this association being resistant to high ionic strength and high-pH conditions [108,109,110,111]. Notably, the abnormal insertion of SOD1 mutants into the ER membrane has been identified as sufficient to induce ER stress, an event that initiates a cascade of cell-specific damage in ALS pathogenesis [110]. A recent study revealed that Lewy bodies are, in fact, a dense mixture of membranous components along with α-synuclein aggregates. The formation of these structures is believed to result from α-synuclein’s dual abilities to aggregate and interact with membranes [112]. This suggests that the combined abilities of aggregation and non-specific membrane interaction may be a fundamental mechanism by which aggregation-prone proteins exert high proteotoxicity, leading to cell death by disrupting membranous organelles, as seen in Lewy bodies. In this context, the less specific a protein’s membrane interaction, the greater its potential membrane toxicity. This observation explains why aggregation-prone proteins are generally toxic, contributing not only to neurodegenerative diseases but also to the aging process, even in organisms as simple as *E. coli* [41,111,112,113]. Nevertheless, the most prominent human diseases caused by protein aggregation are neurodegenerative disorders and cardiac dysfunction, likely because neurons and cardiomyocytes are rarely replaced [114]. This suggests that the ability of protein sequences to transform their structure based on the environment may be a general mechanism underlying proteotoxicity.

Therefore, the capacity for membrane interaction appears to be inherently encoded in all well-folded, non-membrane proteins (I of Figure 2B), although it is typically locked within their native folded structures. When these proteins become destabilized or their structures are disrupted due to genetic factors, such as disease-associated genetic variations, including mutations, truncations, or splicing alterations; as well as due to pathological, environmental, or aging-related factors, such as post-translational modifications (PTMs), disulfide bond reduction, cofactor loss, or oxidative modifications/fragmentation, their conformations become disordered to varying degrees. These disordered states are soluble only in salt-free water (II of Figure 2B). Remarkably, they can undergo distinct transitions depending on their environments. For example, they may transform into membrane-associated proteins (II of Figure 2B) upon encountering membranes or aggregate within cells in the presence of saline buffers (IV of Figure 2B). These transitions are tightly regulated by the intricate interplay between proteins, water, salt ions, and phospholipid molecules. Importantly, in all cases, the transition is associated with the release of bound water molecules into the bulk solvent, a process that can entropically drive the transition.

## 4. Novel Insights into the Hofmeister Effects on Protein Structures, Stability, and Dynamics

Franz Hofmeister’s pioneering work not only introduced the ‘salting-in/salting-out’ principle but also led to the development of the Hofmeister series, which categorizes ions based on their influence on protein stability in aqueous solutions [49,54,115,116,117,118]. As depicted in Figure 3A, ions that promote protein stability are referred to as kosmotropes, while those that decrease stability are known as chaotropes. Strong kosmotropes include Na_2_SO_4_ and Na_2_HPO_4_, whereas GdmCl and NaSCN are strong chaotropes. NaCl, positioned in the middle of the series, is considered neutral. Remarkably, Hofmeister effects are observed across numerous fields, including medicine, biology, chemistry, and industrial science. Despite their widespread relevance, the precise microscopic mechanisms underlying the Hofmeister series remain poorly understood. It is generally thought that these effects result from intricate and specific interactions between ions and proteins, as well as between ions and the water molecules surrounding proteins, which influence protein hydration. However, the extreme complexity of systems involving ions, counterions, solvents, and co-solutes, each playing distinct roles, presents a significant challenge in uncovering the detailed microscopic mechanisms involved. Indeed, the morphological polymorphism of aggregation and amyloid fibrillation appears to further depend on other factors such as cation valency and ionic strength [119,120].

NMR spectroscopy is a powerful tool for characterizing protein conformation, stability, binding, and dynamics [121,122,123,124,125,126], providing valuable insights into the mechanisms by which Hofmeister series ions influence proteins. However, two key challenges arise: (1) At low concentrations, hydrophobic side chains in most proteins remain buried and inaccessible to ions until global unfolding occurs. (2) Upon global unfolding, however, significant changes in protein NMR resonances can result from both conformational changes and ion–protein interactions, making it difficult to distinguish between these two effects. To address these challenges, a folded 39-residue WW4 domain, lacking any cysteine residues (Figure 3B), was recently chosen to evaluate the effects of five salts—neutral NaCl, kosmotropic Na_2_SO_4_ and Na_2_HPO_4_, as well as chaotropic GdmCl and NaSCN—on its conformation, thermal stability, binding, and backbone dynamics at low salt concentrations (≤200 mM) using CD and NMR spectroscopy [124,125].

**Figure 3 ijms-25-12817-f003:**
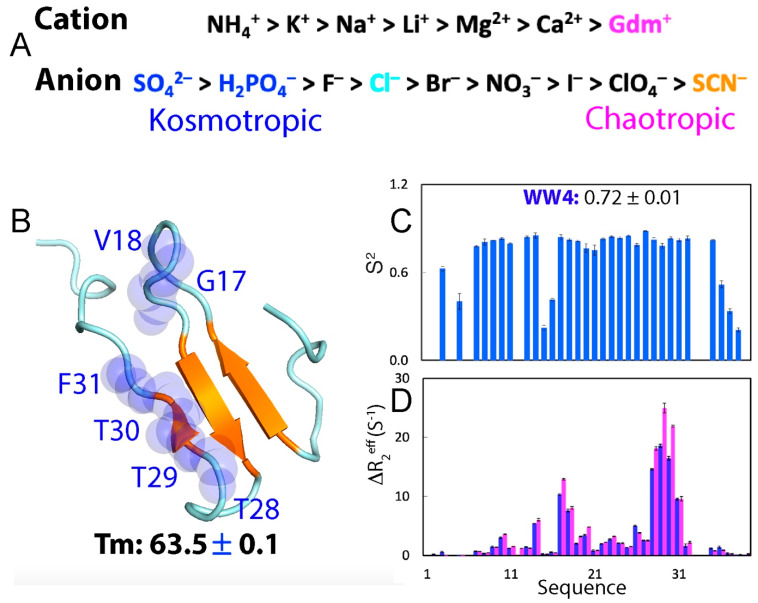
**The Hofmeister effects on the WW4 domain.** (**A**) The Hofmeister series of common cations and anions. (**B**) NMR structure of the WW4 domain with the melting temperature (Tm) labeled, in which 6 residues with significant μs-ms dynamics are displayed in spheres. (**C**) Squared generalized order parameters (S^2^) of WW4. (**D**) Differences in effective transverse relaxation rates (ΔR^2^_eff_) at 80 and 960 Hz, for WW4 domain at 500 MHz (blue) and 800 MHz (purple). Figure 3B was adapted from Figure 7 in Ref. [125].

The WW4 domain from the ubiquitin ligase WWP1 adopts the classic WW fold [126] and offers several significant advantages for studying ion–protein interactions: (1) Unlike larger folded proteins with buried hydrophobic side chains, WW4 consists of a flat, three-stranded β-sheet, making its hydrophilic and hydrophobic residues largely accessible even in the native state (Figure 1B). This accessibility allows WW4 to interact with salt ions even at low concentrations, before global unfolding occurs. As a result, any binding events can be detected by NMR spectroscopy, which is sensitive enough to resolve residue-specific interactions and perturbations over a wide range of affinities, including dissociation constants (Kd) in the millimolar range [120,121,122,123,124,125,126,127,128]. (2) Despite its small size, WW4 exhibits remarkably high thermodynamic stability, undergoing reversible thermal unfolding with a melting temperature (Tm) of 63.5 °C. This Tm is higher than that of many larger folded proteins, such as the 87-residue RRM domain of FUS, which has a Tm of ~52 °C [127], and the 140-residue human profilin 1 (hPFN1), with a Tm of ~56 °C [128]. (3) Regarding ps-ns backbone dynamics, most non-Proline residues in WW4 show order parameters (S^2^) greater than 0.7, with an average value of 0.72, indicating that WW4 is well-folded (Figure 3C). The exceptions are the terminal residues Asn1-Leu5 and Asn36-Ser39, as well as loop residues Arg15-Glu16. Additionally, six residues (Gly17-Val18 and Thr28-Thr30-Phe31) display significant μs-ms backbone dynamics (Figure 3D), undergoing exchanges with a minor conformation that constitutes less than 10% of the population. This minor conformation differs from the major state by less than 0.5 ppm in the backbone ^15^N chemical shift, suggesting that the two exchanging conformations are highly similar [124].

The effects of five salts on the conformation, thermal stability, binding, and backbone dynamics of the WW4 domain were quantified using CD and NMR spectroscopy, as summarized in Figure 4A: (1) Up to a concentration of 200 mM, none of the five salts caused detectable changes to the tertiary structure of WW4. However, they had differential effects on its thermal stability. GdmCl and NaSCN destabilized WW4, reducing the melting temperature (Tm) by ~9.0 °C and ~3.2 °C, respectively [124]. In contrast, Na_2_SO_4_ and Na_2_HPO_4_ stabilized WW4, increasing the Tm by ~5 °C, while NaCl had no effect [125]. These results perfectly align with the classic Hofmeister series ranking (Figure 3A). (2) At the microscopic level, the binding profiles of the five salts to WW4 varied greatly. The cations sodium (Na^+^) and guanidinium (Gdm^+^) showed no detectable binding to the four aliphatic hydrophobic side chains, with only minor binding to amide protons. Among the five anions, only SCN^−^ bound to aliphatic hydrophobic side chains. The four anions exhibited distinct binding profiles to amide protons. For instance, Cl^−^ weakly bound to the Arg27 amide proton, while SCN^−^, SO_4_^2^^−^, and HPO_4_^2^^−^ bound to different sets of residues with varying affinities: SCN^−^ bound to Arg27, Thr28, Thr29, and Thr30; SO_4_^2^^−^ bound to Arg27 and Phe31; and HPO_4_^2^^−^ bound to Trp9, His24, and Asn36. Notably, only SCN^−^ and SO_4_^2^^−^ showed saturable binding profiles, with SO_4_^2^^−^ having a higher affinity than NaSCN. (3) In terms of ^1^⁵N backbone dynamics, all five salts had no significant effect on ps-ns time scale dynamics up to 200 mM. However, their effects on the μs-ms time scale were highly distinct. NaCl and Na_2_SO_4_ had no notable impact, GdmCl reduced μs-ms dynamics, and NaSCN and Na_2_HPO_4_ greatly enhanced them. Remarkably, even at 20 mM, Na_2_HPO_4_ significantly increased μs-ms dynamics. Given that NaCl had no effect, the reduction in μs-ms dynamics by GdmCl is likely attributable to the guanidinium cation. Notably, the chloride anion was inert in terms of all these effects. Overall, the effects of the five salts on WW4 are consistent with previous observations in the field: although the ranking of salt effects on thermodynamic stability is consistent with the Hofmeister series across proteins, the underlying microscopic mechanisms vary significantly [49,50,51,52,53,54,115,116,117,118,119].

These findings prompt us to ask a fundamental question: What underlying mechanisms drive this unique phenomenon for the salts’ effects on protein stability? Protein thermodynamic stability is determined by a complex interplay of several factors. Among these, the hydrophobic effect, electrostatic interactions (such as hydrogen bonds and salt bridges), and the formation of hydration shells through interactions with water play critical roles. Each protein’s stability emerges from a distinct combination of these factors, creating a finely tuned equilibrium. Ultimately, this stability reflects a balance between favorable enthalpic contributions, arising from covalent bonds and non-covalent interactions, and unfavorable entropic contributions, which are largely due to the reduction in conformational entropy that occurs upon folding.

The interaction of salt ions with proteins is primarily influenced by two key factors: charge density and degree of hydration [6,50,115,116,117,118,119]. As shown in Figure 4A, the Gdm^+^ cation and SCN^−^ anion have large ionic volumes but low charge numbers, resulting in low charge densities. Additionally, they exhibit weak hydration, with only 4–6 water molecules surrounding Gdm^+^ and 2–4 water molecules surrounding SCN^−^. Consequently, both ions are thought to disrupt the hydration structure of proteins. In contrast, the SO_4_^2^^−^ and PO_4_^3^^−^ anions have both large ionic volumes and high charge numbers, leading to high charge densities. These anions also demonstrate significant hydration, with 12 water molecules surrounding SO_4_^2^^−^ and 16–20 water molecules around PO_4_^3^^−^. Therefore, both ions are considered to stabilize the hydration structure of proteins. The Cl^−^ anion, meanwhile, has a medium ionic volume and low charge number, resulting in medium charge density, and exhibits a moderate hydration degree with 6–7 water molecules surrounding it.

In this context, the effects of the five salts on the WW4 domain can be explained as follows (Figure 4A,B): NaCl is inert and has no impact on stability, while both the Gdm^+^ cation and Cl^−^ anion exhibit only minor interactions with WW4. The pronounced destabilizing effect of GdmCl can thus be attributed to the Gdm^+^ cation, which significantly disrupts the hydration structure of WW4, leading to reduced μs-ms backbone dynamics. Similarly, NaSCN destabilizes WW4 primarily through the SCN^−^ anion, which breaks down the hydration structure. Additionally, the SCN^−^ anion interacts extensively with hydrophobic side chains, disrupting critical hydrophobic interactions essential for protein thermodynamic stability. This disruption is anticipated to increase μs-ms backbone dynamics, akin to the elevated dynamics observed by NMR in a 37-residue small protein upon pH- and temperature-induced disruption of its hydrophobic core [129,130,131]. In contrast, both SO_4_^2^^−^ and PO_4_^3^^−^ anions exhibit high charge densities and hydration degrees, suggesting that Na_2_SO_4_ and Na_2_HPO_4_ significantly enhance the thermal stability of WW4 by stabilizing its hydration structure. Furthermore, these two anions can interact extensively with WW4 residues but not hydrophobic side chains. Notably, Na_2_SO_4_ did not significantly affect the μs-ms backbone dynamics of WW4, whereas Na_2_HPO_4_ notably increased these dynamics.

In summary, three key factors appear to govern how salts influence protein thermodynamic stability and dynamics: (1) The intrinsic ability of salts to disrupt or stabilize the hydration structure of a protein serves as the primary factor. (2) The capacity of salt ions to bind with protein residues is an additional factor that may vary depending on the specific salt and protein involved. (3) Alterations in protein dynamics likely stem from the interplay between the effects of salts on the protein’s hydration structure and the direct interactions between salts and protein residues. However, changes in dynamics can influence thermodynamic stability positively or negatively, depending on the specific salt and protein, likely through entropy–enthalpy compensation [132,133]. As a result, a simple and general correlation between the thermodynamic stability and the backbone dynamics of WW4 at both ps-ns and μs-ms time scales is not evident.

The most surprising finding is that PO_4_^3^^−^ can significantly enhance μs-ms backbone dynamics, despite stabilizing the protein and lacking the ability to bind to hydrophobic side chains like SCN^−^ [124,125]. This raises the question: What could be the underlying mechanism? As illustrated in Figure 4C, phosphate exists in four ionic states in solution, which continuously exchange due to protonation–deprotonation reactions. Notably, these exchanges typically occur at rates on the μs-ms time scale [134]. The phosphate ion’s strong ability to interact with water molecules and bind to the amide protons of proteins may increase the frequency of exchange processes among its different ionic states within the hydration shell of the WW4 domain. This could trigger μs-ms conformational exchanges in the WW4 domain. Consequently, because WW4 has many conformational states around the native state, the phosphate anion acts to facilitate the transition from a less stable state to the more stable state, which are separated by relatively low energy barriers (Figure 4B).

However, in the context of amyotrophic lateral sclerosis (ALS)-associated mutants of C71G-PFN1 and nascent hSOD1, which exhibit both folded and unfolded states separated by larger energy barriers, phosphate alone is insufficient to induce folding. As a result, covalently linked polyphosphates, such as diphosphate, triphosphate, or ATP or ADP, are required to facilitate the folding transition [128]. This mechanism may help explain the richness of various phosphate-containing molecules in intracellular environments, where, in addition to approximately 10 mM of free phosphate, there are around 45 mM of phosphocreatine, 3.5 mM of hexose phosphate, and 3–12 mM of ATP [47,48]. While further studies are needed to explore this hypothesis, these molecules may play a crucial role in enhancing protein stability and promoting the folding of a diverse range of proteins.

## 5. ATP Effectively Antagonizes the Crowding-Induced Destabilization

Adenosine triphosphate (ATP) is the universal energy currency in all living cells, driving numerous biological processes through the hydrolysis of its high-energy bonds (I of Figure 5A), which are thermodynamically favorable but kinetically controlled [47,48]. Its central role includes fueling metabolic reactions, such as glycolysis and oxidative phosphorylation, which are essential for producing the energy required for cellular functions. ATP also plays a pivotal role in signal transduction, where it acts as a substrate for kinases in phosphorylation reactions, thereby regulating various cellular pathways. Moreover, ATP is crucial for active transport mechanisms, including the functioning of ATP-binding cassette (ABC) transporters and sodium–potassium pumps, which maintain ion gradients across cell membranes. ATP is also a precursor for nucleic acid synthesis and is involved in the synthesis of RNA and DNA. Additionally, ATP serves as a signaling molecule in both intracellular and extracellular environments, influencing processes such as muscle contraction and neurotransmission [47,48].

Strikingly, a recent study proposed that ATP may have originated from prebiotic chemical processes even before the polymerization of RNA, DNA, and proteins or the evolution of genetically encoded macromolecular machines. Notably, ATP has been shown to form efficiently only in slightly acidic, unsalted water via the phosphorylation of ADP by acetyl phosphate (AcP), a conserved intermediate linking thioester and phosphate metabolism [96]. Mysteriously, while ATP-dependent enzymes and proteins in modern cells operate at micromolar concentrations, intracellular ATP levels are much higher, typically ranging from 2 to 12 mM depending on the cell type [47,48]. For example, despite being metabolically inactive, the vertebrate lens maintains ATP concentrations between 3 and 7 mM [19,135]. Moreover, although viruses lack ATP [47,48], recent findings reveal that ATP modulates the liquid–liquid phase separation of the SARS-CoV-2 nucleocapsid (N) protein, a critical process in viral infection and replication [136,137,138]. The ubiquitous presence and conservation of ATP across all biological systems suggest its early emergence in biochemical evolution, potentially influencing fundamental biological processes beyond its traditional roles and shaping essential biological functions and the genome–proteome interface throughout evolutionary history [96,139].

Indeed, ATP has been found to function as a biological hydrotrope at concentrations exceeding 5 mM, dissolving LLPS and protein aggregates or fibrils [140,141]. This newly discovered role is attributed to ATP’s amphiphilic nature, comprising a hydrophobic adenine and a hydrophilic triphosphate moiety (II of Figure 5A). Specifically, the aromatic purine ring of ATP is thought to cluster over hydrophobic regions of protein droplets or aggregates, while its triphosphate group interacts strongly with bulk water, thereby facilitating the dissolution of LLPS and aggregates.

Furthermore, NMR analysis has provided residue-specific insights into ATP’s interactions with IDRs of FUS, revealing that ATP can biphasically modulate the RGG-rich domain, which lacks large hydrophobic residues and cannot undergo phase separation independently. At low ATP concentrations, it induces LLPS, while at higher concentrations, it dissolves the phase separation [73,142]. This suggests that ATP acts as a bivalent binder, selectively targeting Arg/Lys residues (III of Figure 5A). Additionally, ATP has been shown to bind to conformation-specific pockets in folded proteins, inhibiting amyloid fibrillation either thermodynamically or kinetically [123,143,144]. More recently, even without significant binding or direct effects on the thermodynamic stability at low protein concentrations, ATP has been found to antagonize crowding-induced destabilization of human lens γ-crystallin at extremely high protein concentrations [145,146,147]. ATP appears to exert these effects by functioning as a hydration mediator (IV of Figure 5A). On the other hand, high-resolution microwave dielectric spectroscopy has revealed that around ATP’s triphosphate chain, not only is there constrained water with a dielectric relaxation time longer than that of bulk water, but also an “anomalous” layer of hyper-mobile water with a relaxation time shorter than bulk water [148]. This unique capacity of ATP in increase dynamics of the hydration shell is likely attributed to its triphosphate group, whose different ionic states are continuously exchanging on an s-ms time scale, as proposed for the phosphate group (Figure 4C).

The vertebrate lens is extremely crowded with proteins, reaching concentrations of 200–400 mg/mL in the human lens and exceeding 1000 mg/mL in fish lenses [135,149]. The predominant proteins in the lens are crystallins, which are categorized into three main types—α, β, and γ crystallins—collectively comprising about 90% of the total lens proteins. In the human lens, α-crystallins account for roughly 28%, β-crystallins for 43%, and γ-crystallins for another 28% [149,150,151]. Functionally, α-crystallins act as molecular chaperones, while β- and γ-crystallins belong to the βγ-crystallin superfamily, serving primarily structural roles. It has been previously shown that α- and β-crystallins can form polydisperse oligomers, whereas γ-crystallins typically exist in a monomeric form in their native state. Notably, because the lens lacks mechanisms for protein repair or recycling, crystallins must remain functional throughout the entire lifespan of an individual. As such, maintaining the solubility and stability of lens crystallins is critical to their physiological role [135,145,146,147,149,150,151,152,153,154,155,156,157,158,159]. Importantly, genetic mutations or post-translational modifications can lead to the aggregation of crystallins, potentially resulting in cataract formation—an eye disease identified as a priority by the World Health Organization and responsible for 48% of global blindness.

The impact of molecular crowding on protein stability is a fundamental topic in protein science. It has been suggested that the crowded environment within cells, where macromolecules like proteins, nucleic acids, and polysaccharides occupy a significant portion of the available volume, restricts the space available for protein unfolding. This “excluded volume effect” is thought to stabilize the native protein state by limiting the conformational freedom necessary for unfolding [160,161,162,163,164]. However, subsequent studies revealed that macromolecular crowding, both in vitro and in vivo, can also destabilize proteins due to non-specific intermolecular interactions. Thus, the effect of crowding on protein stability is highly context-dependent, influenced by factors such as the specific protein and the conditions within the cellular environment [162,163,164].

The βγ-crystallins have been evolutionarily optimized for their high solubility, stability, and refractive properties, which are essential for forming the eye lens and its characteristics of transparency and a refractive index gradient. In the cortex of the human lens, the 178-residue γS-crystallin is the most abundant structural protein. Interestingly, despite γS-crystallin exhibiting weak attractive intermolecular interactions, it remains predominantly monomeric even at very high concentrations. Studies have shown that these attractive interactions are independent of salt concentration, leading to the proposal that this behavior may be attributed to its unique hydration shell [165,166,167]. The three-dimensional structure of human γS-crystallin, determined by NMR spectroscopy [165], reveals the characteristic four Greek key fold typical of the βγ-crystallin superfamily, organized into distinct N- and C-terminal domains (Figure 5B). Additionally, the packing interactions between these N- and C-domains are thought to play a crucial role in maintaining the protein’s high solubility and stability, even in the highly crowded and densely packed environment of the lens.

To date, four cataract-causing mutations have been identified within the N-domain of γS-crystallin: G18V, D26G, S39C, and V42M (Figure 5B) [145,146,147,166]. Among these, the G18V mutation is associated with childhood-onset cortical cataracts. Interestingly, biophysical studies have shown that the G18V mutant exhibits only a slight reduction in thermodynamic stability compared to the wild-type (WT) protein. However, unlike the WT, the G18V mutant is highly prone to aggregation even at temperatures well below its unfolding threshold [153,167,168]. Structural studies using NMR revealed only localized changes in the G18V γS-crystallin compared to the WT, suggesting that the mutation’s aggregation mechanism is more complex than simple mutation-induced denaturation [165]. Notably, recent research has demonstrated that WT γS-crystallin possesses a remarkably robust hydration shell, which allows it to resist the crowding effects even at protein concentrations as high as 400 mg/mL. In contrast, the G18V mutation disrupts this hydration shell, rendering γS-crystallin more susceptible to aggregation, even at low concentrations and temperatures far below its unfolding temperature [153,167,168].

Therefore, understanding the mechanisms that enable crystallins to maintain high solubility and stability throughout life in the crowded environment of the lens is of both fundamental and therapeutic importance. Despite extensive research, several critical aspects remain unresolved. In particular, why does the metabolically inactive lens maintain such high ATP concentrations (3–7 mM), levels that exceed those found in neurons?

Recent studies aimed to investigate whether ATP influences the association, stability, and conformation of both the wild-type (WT) and three cataract-associated mutant forms of human γS-crystallin. These studies characterized the effects of ATP at protein concentrations ranging from 0.1 to 5 mM (~100 mg/mL) using techniques like NMR spectroscopy, dynamic light scattering (DLS), and thermal unfolding [145,146,147]. Key findings include the following:WT γS-crystallin remains monomeric even at concentrations around 100 mg/mL in buffer containing 150 mM NaCl, mimicking in vivo conditions. At a low protein concentration (0.1 mM), WT is highly stable with a melting temperature (Tm) of 71 °C. However, self-crowding reduces thermal stability, with the Tm dropping to 55.5 °C at 5 mM (Figure 5C).The four cataract-associated mutations (G18V, D26G, S39C, and V42M) significantly increase the tendency for aggregation. Notably, the G18V mutant begins to precipitate at concentrations above 1 mM, while WT only starts forming a gel-like state at concentrations above 5 mM [123]. The other three mutants precipitate at concentrations exceeding 1.5 mM. Despite these aggregation tendencies, at low concentrations (0.1 mM), the mutations only slightly reduce thermal stability, with Tm values of 67.0 °C, 69.0 °C, 70.5 °C, and 68 °C for G18V, D26G, S39C, and V42M, respectively. However, with increasing protein concentrations (self-crowding), the Tm values drop sharply, with G18V falling to 50.5 °C at 1 mM, and D26G, S39C, and V42M decreasing to 50.0 °C, 57.0 °C, and 51.0 °C, respectively, at 1.5 mM (Figure 5C).ATP shows no significant binding to WT or the mutant γS-crystallins, nor does it affect their conformation, even at a high molar ratio of 1:200 (γS-crystallin). ATP only induces minor changes in HSQC peaks, similar to those caused by protein dilution.ATP also has no detectable effect on αB-crystallin as well as its interactions with WT or mutant γS-crystallins.Unexpectedly, ATP differentially antagonizes the crowding-induced destabilization of WT and the mutant proteins, even at a molar ratio of 1:1. In WT, the Tm increases from 55.5 °C to 62 °C at 5 mM. Surprisingly, ATP no longer prevents the crowding-induced destabilization of the G18V mutant at 1 mM. For the D26G, S39C, and V42M mutants, ATP still partially antagonizes the crowding-induced destabilization, raising the Tm values from 50.0 °C, 57.0 °C, and 51.0 °C to 53.5 °C, 59.5 °C, and 58.5 °C, respectively, at 1.5 mM (Figure 5C).

These findings suggest that ATP can effectively antagonize the crowding-induced destabilization of WT γS-crystallin, which has a robust hydration shell, but not the G18V mutant, where the hydration shell is disrupted [146,152]. This implies that ATP most likely antagonizes crowding-induced destabilization by interacting directly with the unique hydration shell of human γS-crystallin. The acceleration of the crowding-induced destabilization by cataract-causing mutations appears to result not only from the well-known structural perturbations but also from alterations in the protein’s hydration shell. This hydration shell, which plays a central role in protein folding and aggregation, has not been fully appreciated until now.

In light of the current understanding of protein folding and aggregation, a novel mechanism has been proposed to explain how ATP antagonizes the crowding-induced destabilization of γS-crystallin, even at a 1:1 molar ratio. As illustrated in Figure 5D, at low concentrations, γS-crystallin possesses a robust hydration shell that plays a critical role in maintaining its stability (I of Figure 5D). However, in the extremely crowded environment of the lens, even the hydration shell of WT γS-crystallin becomes distorted, with this effect being far more pronounced in cataract-causing mutants (II of Figure 5D). ATP appears to possess the ability to effectively counteract the crowding-induced destabilization by directly interacting with the intact hydration shell of WT γS-crystallin, thereby enhancing its ability to resist the crowding effect. In contrast, the G18V mutation significantly disrupts the intrinsic hydration shell, leaving no possibility for ATP to interact, which explains why ATP loses its stabilizing capacity for this mutation. Thus, in addition to the chaperone activity provided by α-crystallin, ATP at millimolar concentrations is also essential to antagonize the destabilization caused by the extreme crowding of crystallins in the eye lens through this newly identified mechanism. This insight also helps explain the connection between declining ATP levels with aging [169,170] and age-related cataractogenesis, a condition that affects virtually all older individuals. Consequently, restoring normal ATP concentrations in the eye lens could offer a promising therapeutic strategy for treating aggregation-related eye diseases like cataracts.

## 6. ATP Induces Protein Folding with the Highest Capacity

The majority of proteins that require folding from the unfolded state (U) to the folded state (F) for their function are only marginally stable [14,15,171,172,173,174,175]. As a result, genetic mutations and some environmental stresses can easily destabilize these proteins, leading to misfolding and aggregation in cells, particularly in crowded environments [170] and under high salt concentrations [15]. This destabilization is a common pathological feature of aging and neurodegenerative diseases. Currently, it is widely accepted that cells primarily address protein folding and misfolding/aggregation through supramolecular machinery that is energetically powered by ATP [176,177,178]. However, it remains unclear whether ATP can directly influence the protein folding equilibrium itself, an essential process at the heart of protein homeostasis.

Amyotrophic lateral sclerosis (ALS) is the most common motor neuron disease, first described in 1869, yet its underlying mechanisms remain largely elusive [179]. Approximately 90% of ALS cases are sporadic (SALS), while 10% are classified as familial ALS (FALS). In 1993, human CuZn-superoxide dismutase 1 (hSOD1) was identified as the first causative gene linked to the most common form of FALS [180]. To date, more than 180 mutations have been identified in the 153-residue SOD1 protein [181]. Additionally, misfolding and aggregation of wild-type (WT) hSOD1 have also been extensively associated with SALS [71,180,181,182,183]. Subsequently, several mutations of human profilin 1 (hPFN1), a 140-residue protein that plays a crucial role in regulating actin polymerization physiologically [184], have been linked to familial ALS (FALS). The protein adopts a seven-stranded antiparallel β-sheet, which is sandwiched between α-helices at the N- and C-termini on one side and three smaller helical regions on the opposite face (I of Figure 6A). Its C71G mutation with Cys71 replaced by Gly is the most toxic and prone to misfolding and aggregation [107]. As a result, all attempts to determine the crystal structure of this mutant have been unsuccessful [185]. Remarkably, NMR studies have shown that C71G-hPFN1 exists in a dynamic equilibrium between unfolded and folded states, which is characterized by two distinct sets of HSQC peaks (I of Figure 6B). This observation indicates the presence of an energy barrier separating the two states [72,128]. The populations of these states were quantified by NMR, revealing that the folded state comprises approximately 55.2% while the unfolded state accounts for about 44.8%, with an average exchange rate of ~11.7 Hz (~85.5 milliseconds) (II of Figure 6A).

To evaluate the effects of salt ions on the folding of nascent hSOD1 and C71G PFN1, a comprehensive series of common cations and anions were examined using NMR spectroscopy. Among the 12 cations tested, only Zn^2^^+^ and Fe^2^^+^ were capable of specifically inducing the folding of nascent hSOD1, resulting in a coexistence of folded and unfolded states; the remaining ions led to precipitation at high concentrations. In fact, Zn^2^^+^ is a well-recognized cofactor essential for initiating the proper folding of hSOD1, while Fe^2^^+^ served as the cofactor in the ancestral Fe-SOD1, which shared the same fold as CuZn-SOD1. However, following the Great Oxidation Event, the environmental availability of Fe^2^^+^ declined, leading to its disappearance in modern organisms. In the case of C71G-hPFN1, a large spectrum of cations and anions, including phosphate, were tested for their ability to promote the conversion of the unfolded state into the folded state. However, none of them succeeded; instead, all led to protein aggregation. These findings suggest that, except for specific cofactors, all salt ions exert non-specific electrostatic screening effects, which in turn promote significant aggregation of nascent hSOD1 and C71G-PFN1.

Very unexpectedly, ATP has been discovered to completely convert C71G-hPFN1 into the folded population even at a ratio of 1:2 (I of Figure 6B) and to transform the nascent hSOD1 into an equilibrium of the folded and unfolded states at 1:8 but without any specific binding to two proteins [128]. To decode the underlying mechanism, the effects of 12 related compounds (Figure 6C) were further visualized by systematic NMR titrations. Notably, ATP and triphosphate were both able to completely convert the unfolded state of C71G-hPFN1 into its folded state at a 1:2 ratio (II of Figure 6B). In contrast, TMAO, a well-known molecule for promoting protein folding [186,187,188], showed no detectable induction, even at high ratios up to 1:2000, where C71G-hPFN1 instead precipitated. These results suggest that ATP and phosphate exhibits the strongest folding-inducing capacity observed to date.

Although phosphate lacks the ability to induce folding, ADP and diphosphate also exhibit the same folding-inducing capacity, though weaker than that of ATP. The hierarchy of folding-inducing ability is decoded as follows: ATP = ATPP = PPP > ADP = AMP-PNP = AMP-PCP = PP (Figure 6D), while AMP, adenosine, phosphate, TMAO, and NaCl show no inducing effect. Interestingly, results from ATP analogs revealed that very specific structural features are necessary for inducing protein folding. This capacity is not only determined by the number of phosphate groups but also by the atoms linking them. While phosphate cannot induce folding and diphosphate has only weak activity, triphosphate displays the same folding capacity as ATP. However, Adenosine 5′-(pentahydrogen tetraphosphate) (ATPP) shows the same effectiveness as ATP. Moreover, the ability to induce folding also depends on the atoms connecting the beta and gamma phosphates. When the oxygen atom linking these phosphates is replaced with a carbon atom in methyleneadenosine 5-triphosphate (AMP-PCP) or a nitrogen atom in adenylylimidodiphosphate (AMP-PNP), their folding capacity is reduced to the level of ADP. Most intriguingly, while ATP and ADP do not induce aggregation of C71G-PFN1 even at 20 mM, triphosphate, diphosphate, ATPP, AMP-PCP, and AMP-PNP all trigger aggregation at very low concentrations [128].

So, what could be the mechanism for ATP and triphosphate to induce folding at such a high effectiveness for both hSOD1 and C71G-PFN1, which have unrelated structures and functions? It appears to be very unlikely that the mechanism is just a simple volume exclusion or electrostatic screening effect. At such a low concentration for ATP and triphosphate to induce folding, the volume-excluding mechanism proposed for TMAO at concentrations > M [172,186,187,188] does not operate at all. The possibility that the folding-inducing capacity arises only from electrostatic screening can also be ruled out, as the ionic strength of sodium triphosphate (PPP) is only 15 times greater than that of sodium chloride. Nevertheless, sodium chloride at 10 mM, with a much higher ionic strength than PPP at 0.1 mM, fails to induce folding but instead triggers the aggregation of C71G-hPFN1 [128].

The most plausible explanation for ATP and triphosphate to induce folding at the highest efficiency is likely their unique and strong ability to interact with protein hydration, which is rooted in phosphate (Figure 4C). Previous NMR studies revealed that ATP and triphosphate exhibit a high capacity to mediate protein hydration, even without strong or specific binding interactions [73,142]. Recent NMR and molecular dynamics (MD) simulation studies indicate that ATP generally has weak and non-specific interactions on folded proteins, even at very high ATP-to-protein ratios. However, these interactions are sufficient to mediate the hydration shell of proteins [189]. On the other hand, it has been suggested that hydrogen bonding with water molecules, particularly involving protein backbone atoms, plays a critical role in protein folding [171,172]. The unfolded state is stabilized when key backbone atoms are hydrogen-bonded to water molecules, whereas the folded state is favored when these atoms form intramolecular hydrogen bonds.

In this context, ATP and triphosphate appear to primarily facilitate protein folding by displacing water molecules that form hydrogen bonds with some key residues of the unfolded state. In the unfolded state (I of Figure 7A), the protein is heavily hydrated, with key residues interacting with water molecules through hydrogen bonds, which creates a kinetic barrier to folding. The introduction of ATP or triphosphate disrupts these water–protein interactions, loosening the water molecules and initiating the folding process (II of Figure 7A). The release of water molecules from the bound state to bulk water during folding also increases entropy, thus energetically driving folding. However, due to the exposure of the hydrophobic patches in such a protein with mutated deficits or highly dynamic tertiary packing, excessive amounts of triphosphate can lead to aggregation, caused by strong electrostatic screening effects. In contrast, ATP not only promotes folding through its triphosphate group but also enhances thermodynamic stability. This may be achieved through dynamic interaction of the aromatic base ring of ATP with the exposed hydrophobic patches or filling structural cavities, thus protecting these patches from exposure to bulk water (III of Figure 7A) [128]. This mechanism underscores that ATP and triphosphate enhance the folding propensity already encoded in the protein sequence. Indeed, ATP was unable to induce folding of intrinsically disordered domains, such as the prion-like domain of TDP-43 [190] and the RGG-rich domain of FUS [73], even at a 1:400 ratio, as these sequences inherently lack the capacity to form folded domains.

## 7. ATP and Nucleic Acids Competitively Modulate LLPS of Arg/Lys-Containing IDRs

LLPS is now widely recognized as a common mechanism driving the formation of various membrane-less organelles (MLOs) and cellular condensates. Among these MLOs are nucleoli, Cajal bodies, nuclear speckles, paraspeckles, histone-locus bodies, nuclear gems, and promyelocytic leukemia (PML) bodies within the nucleus, as well as P-bodies, stress granules (SGs), and germ granules in the cytoplasm [191,192,193,194]. Earlier studies primarily focused on the phase separation of well-structured proteins, such as lysozyme, showing that this process only occurs at high concentrations (>mM) [195,196,197,198,199,200]. In contrast, proteins involved in the formation of membrane-less organelles typically contain large IDRs, allowing phase separation at much lower concentrations (~μM) [201,202,203,204,205,206]. The key to this phenomenon lies in the multivalency of binding sites within IDR-rich proteins, which enables them to interact simultaneously with multiple copies of themselves (homotypic phase separation) or with other biomolecules (heterotypic phase separation). This leads to LLPS, resulting in the separation of a homogeneous solution into two distinct coexisting phases: a dense phase and a dilute phase [201,202,203,204]. LLPS is dynamic and reversible, with molecules continuously exchanging between the dense and dilute phases. Despite this exchange, the concentration of molecules in the dense phase is often more than 50 times higher than in the dilute phase.

The major mechanisms for phase separation [195,196,197,198,199,200,207,208,209] are often framed by the Flory–Huggins theory [210,211,212,213,214]. Initially developed for polymer solutions [210,211,212], this theory was later extended by Scott and Tompa to describe polymer mixtures in a common solvent [213,214]. However, its applicability to biological LLPS involving IDRs remains a matter of ongoing debate [204]. One major critique is that even though the Flory–Huggins model fits the experimental phase diagram satisfactorily, it offers little insight into the underlying mechanisms of phase separation and lacks predictive capability [204]. According to the Flory–Huggins model, phase separation is driven by repulsive interactions between distinct macromolecules within an inert solvent. However, this framework may not fully capture the behavior of IDRs, which are highly polar or charged and lack a stable three-dimensional structure. Due to their extended conformation and dynamic nature, nearly all regions of an IDR are exposed to the solvent. This suggests that the hydration properties of intrinsically disordered protein regions (IDPRs) differ significantly from those of well-folded, globular proteins.

Indeed, numerous studies using nuclear magnetic resonance (NMR) relaxation and other biophysical methods have shown that IDRs can bind significantly larger amounts of water than globular proteins. Furthermore, the energy barriers governing water motion in IDPs are notably distinct from those observed in globular proteins [78,79,80,81,82]. Since the hydration shell is involved in various molecular interactions, water cannot simply be treated as an inert solvent in IDR-mediated LLPS. Despite its critical role, the contribution of water to biological phase separation and the formation of proteinaceous MLOs is still largely underappreciated. Currently, it is widely accepted that weak, multivalent interactions help to counterbalance the loss of conformational entropy during phase separation in IDRs. Nevertheless, the strong affinity of IDRs for water molecules suggests that the release of relatively constrained water molecules into the bulk water may also play a significant role as a driving force in biological LLPS.

ATP has been shown to modulate LLPS of full-length proteins and their IDRs in a biphasic manner, including those of FUS [73,142], TDP-43 [190,215], and nucleocapsid (N) proteins [136,137,138,216,217]. For full-length proteins, LLPS is not only involved in interactions within the IDRs but also in additional interactions from their folded domains. These include the oligomerization of the hydrophobic region in the prion-like domain of TDP-43 and the dimerization of the C-terminal domain of the SARS-CoV-2 N protein. As a result, ATP’s regulation of LLPS in these proteins is more complex due to the involvement of multiple structural elements. Here, in order to focus on elucidating how ATP biphasically modulates LLPS specifically within IDRs, only the intrinsically disordered C-terminal domain (CTD) of FUS (residues 371–526), which is rich in arginine–glycine–glycine (RGG) repeats, was reviewed. This region lacks large hydrophobic residues and cannot undergo phase separation on its own under various buffer conditions, even at concentrations as high as 1 mM [73,142].

Despite this, ATP has been found to induce LLPS at low concentrations but disrupt it at higher concentrations. When using NMR to examine residue-specific interactions, ATP was found to modulate LLPS of the FUS CTD, likely by acting as a bivalent binder, specifically interacting with Arg/Lys residues (III of Figure 5D). This mechanism was further validated in studies of the prion-like domain of TDP-43, which contains only six arginine and two lysine residues, and exhibited a biphasic response to ATP [190,215]. Similar behavior has also been observed in the RGG-rich IDR of cold-inducible RNA-binding protein [218]. In this context, a mechanism has been proposed for how ATP biphasically modulates the LLPS of Arg-/Lys-containing IDRs using the FUS C-terminal domain (CTD) as an example (Figure 7B). The FUS CTD, being intrinsically disordered and rich in positively charged Arg and Lys residues, is highly hydrated (I of Figure 7B). When ATP is added at low concentrations, it acts as a bivalent binder, crosslinking FUS CTD molecules through interactions with Arg/Lys residues. This binding also perturbs the hydration shell, leading to condensation of FUS CTD molecules and the release of bound water molecules into the bulk water. The release of these water molecules is expected to energetically favor the formation of large, dynamic complexes, which manifest as phase-separated liquid droplets (II of Figure 7B). At higher ATP concentrations, however, excessive ATP binding occurs. This overwhelming interaction disrupts the formation of these large, dynamic complexes, ultimately leading to the dissolution of the phase-separated droplets (III of Figure 7B).

Strikingly, studies combining semiempirical quantum mechanical (SQM) methods, mean-field theory, and coarse-grained molecular dynamics (CGMD) simulations have demonstrated that ATP can indeed act as a bivalent or even trivalent binder, capable of both enhancing and inhibiting the phase separation of FUS [219]. More recently, a comprehensive investigation using a combination of experimental techniques and simulations uncovered unique properties and mechanisms underlying ATP-induced phase separation in basic intrinsically disordered proteins (bIDPs). Briefly, ATP functions as a bridging agent that crosslinks bIDP chains, forming mesh-like networks that manifest as liquid-like droplets. These ATP-induced droplets display unusual physicochemical properties, including the accumulation of very high ATP concentrations within the droplets, rapid droplet fusion, low interfacial tension, and elevated viscosity. As a result, these droplets exhibit extreme shear-thinning behavior, where viscosity decreases under applied stress, making them highly dynamic and adaptable [220,221].

Like ATP, nucleic acids have also been shown to biphasically modulate LLPS, and most notably, ATP and nucleic acids compete for binding to Arg-/Lys-rich regions. However, nucleic acids differ from ATP in their structure, being composed of multiple covalently linked nucleotides, which allows them to establish multivalent interactions with Arg/Lys residues in a length-dependent manner. This makes nucleic acids act similarly to a polymer of ATP. Nevertheless, ATP possesses a unique triphosphate group that gives it a strong capacity to interact with water molecules and Arg side chains. These differences may influence how ATP and nucleic acids compete for binding to Arg/Lys residues and their respective roles in modulating LLPS [139].

This discovery has significant implications for understanding the regulation of fundamental cellular processes. Since most IDRs are expected to contain multiple Arg/Lys residues, many IDRs, like the 40-residue disordered Nogo-40 (which contains two Arg and three Lys residues), are likely capable of binding both ATP and nucleic acids to drive LLPS, even if they do not have a known function associated with these interactions [222]. In this context, a large portion of the human proteome, approximately half of which consists of IDRs, may be predisposed to undergo LLPS through multivalent interactions with nucleic acids.

However, while nucleic acids can achieve a high binding affinity to IDRs through multivalent but discrete interactions with Arg/Lys residues, this binding appears to be vulnerable to displacement by ATP. Since discrete nucleic acid binding events appear to be largely independent and thus can be simultaneously displaced by ATP, it may explain why only a limited number of MLOs are observed in cells. At high intracellular concentrations, ATP may play a previously unrecognized role in inhibiting LLPS in many Arg-/Lys-containing IDRs. Even in MLOs that are already formed, such as stress granules (SGs), ATP is likely essential for maintaining their dynamic and reversible nature, preventing them from transitioning into irreversible aggregates, which are linked to a growing number of human diseases, including all known neurodegenerative disorders. This suggests that ATP’s modulation of IDR–nucleic acid complexes is critical for cellular homeostasis and may serve to protect against pathological conditions associated with excessive phase separation and aggregation.

## 8. The Roles of ATP and Polyphosphates in the Origin of Protocells

The origin of life on Earth is thought to have resulted from a series of geochemical events, where simple prebiotic inorganic molecules interacted to eventually form biopolymers, leading to the development of protocells [223,224,225,226,227,228]. Three fundamental elements were crucial to this process: (1) an energy source (biological fuel) to power metabolic and biochemical reactions, (2) molecules capable of encoding, storing, and transmitting genetic information for reproduction, and (3) functional molecules that catalyze biochemical reactions and serve structural roles. These components operated within compartmentalized aqueous environments, facilitating essential processes such as metabolism and replication. Nature seems to have solved the first two requirements by selecting phosphorus as a fundamental building block across all living organisms. For the third challenge, proteins were engineered as the primary functional molecules, possessing an extreme diversity of three-dimensional structures, including folded domains, IDRs with varying properties, and membrane proteins.

Protocells, also known as primitive cells, are thought to represent the earliest life-like entities, exhibiting key functions such as proto-metabolism, compartmentalization, and replication [225,226,227,228,229]. Three main types of protocells have been proposed: membrane-free coacervate droplets, lipid vesicles, and hybrid protocells that arise from a combination of these models [228,229,230,231,232]. Among these, the coacervate droplet model has attracted particular attention due to its basis in LLPS, which allows the spontaneous sequestration and concentration of polyelectrolytes and biomolecules [233,234,235,236]. Remarkably, coacervate droplets can mimic several critical cellular functions, including biomolecular crowding and intercellular communication, providing valuable insights into early cell-like behavior.

On the early Earth, where conditions were extreme, synthetic organic polymers or modern biomacromolecules [235,236] likely did not exist. For example, the intensity of terrestrial radioactivity on the primordial Earth is estimated to have been up to 4000 times higher than it is today. Ionizing radiation from both cosmic rays and intrinsic radioisotopes might have played a dual role: while potentially driving key prebiotic chemical reactions, it could also have had harmful effects on emerging biomolecules, such as nucleic acids, which are essential for life [237,238,239,240]. However, inorganic polyphosphate (polyP)—a linear, negatively charged polymer made up of tens to hundreds of orthophosphate (Pi) units linked by high-energy phosphoanhydride bonds—emerged long before biological molecules [237,238]. This makes polyP a plausible energy source and phosphate donor for ATP and polynucleotide synthesis during the early stages of life’s evolution [241,242,243,244]. It is suggested that polyP played a crucial role in the origin and survival of life by providing a flexible, polyanionic scaffold, facilitating the assembly and orientation of biomolecules in prebiotic cells. Remarkably, ATP is also believed to have emerged in the early stages of biochemical evolution [96]. The highest ATP yield was observed in highly pure, unsalted water (such as HPLC-grade water) at an optimal pH of approximately 5.5–6. In contrast, the yield significantly decreased when salts were introduced or when the pH deviated from this range. This specific aqueous environment aligns with conditions that can solubilize otherwise insoluble proteins, and it mirrors the likely chemical environment of prebiotic oceans or early water bodies.

LLPS has been increasingly proposed as a mechanism for concentrating and compartmentalizing biological molecules even at extremely dilute concentrations in aqueous environments [245]. In light of this, here, a mechanism is proposed to emphasize the role of ATP in the origin of protocells. In the early stages of biochemical evolution (I of Figure 7C), the water body was likely highly unsalted and slightly acidic, conditions under which primordial peptides and proteins were soluble. The abundant presence of polyphosphate and triphosphate could have driven phase separation of primordial peptides and proteins with cationic residues, thereby concentrating these rare biomolecules (II of Figure 7C).

Although polyphosphate and triphosphate have the ability to induce folding and enhance the solubility of well-folded proteins, they can also promote the aggregation of partially folded and disordered proteins with exposed hydrophobic regions, such as C71G-PFN1 and nascent hSOD1. This observation may explain why today, polyphosphates mostly only present at high concentrations within single-celled organisms, primarily acting as primordial protein chaperones [246,247,248,249,250]. In these organisms, most proteins are well folded, with the low increase in protein aggregation potentially induced by polyphosphate/triphosphate.

As ATP emerged and accumulated, its binding to primordial peptides and proteins could have played a key role in driving their phase separation, forming membrane-less compartments (III of Figure 7C). ATP may have also protected these proteins from aggregation even with the rising salt concentrations in the water bodies. As nucleic acids later appeared, they likely bound to primordial peptides and proteins, further promoting phase separation and leading to the formation of dynamic protein–nucleic-acid complexes. Finally, these phase-separated droplets might have become enclosed by primitive membranes, leading to the formation of early cells (IV of Figure 7C).

Within these primordial cells, mechanisms for self-replication of proteins and nucleic acids may have evolved and become established. Over time, with the extensive sampling of protein sequences, some proteins began folding into folded structures as enhanced by ATP, while others remained intrinsically disordered. Under ATP’s influence, protein–nucleic-acid complexes likely evolved into two distinct forms: well-folded and tight complexes, exemplified by the ribosome, and dynamic droplets that would form membrane-less organelles (V of Figure 7C). Eventually, cellular compartments became further separated by membranes, primarily composed of phospholipids, which feature both hydrophilic and hydrophobic regions [251,252,253,254].

## 9. ATP Acquires the Capacity to Site-Specifically Target the Hydration of Biomolecules

Phosphorus has been selected as a key element for constructing three of the four essential biomolecules of life: ATP, nucleic acids, and phospholipids [255,256,257,258,259,260,261,262]. Its versatility is indispensable for life, also playing extensive roles in signal transduction through phosphorylation; mineralization, with phosphate in hydroxyapatite comprising ~60% of bone mass to ensure skeletal strength; and buffering, stabilizing intracellular pH at ~7.4. Additionally, many other phosphate-containing compounds exist, such as coenzymes (e.g., NAD+/NADP+, FAD, and CoA), phosphosugars (e.g., fructose-1,6-bisphosphate), inositol phosphates, and energy-related molecules like phosphocreatine and phosphoenolpyruvate [47,48]. The prevalence of phosphate-based molecules has led to the adage, “Where there’s life, there’s phosphorus.” A key question thus arises: Why does nature chose phosphates? [259,260,261,262]. It is widely believed that phosphorus was selected due to its many unique chemical properties. These include its ability to store and release energy efficiently, its balance between stability and reactivity, and its versatility in supporting a wide range of biochemical functions essential for life.

However, the unexpected findings reviewed here highlight a previously underappreciated characteristic of phosphate: its strong capacity for self-hydration and its role in regulating the hydration dynamics of other biomacromolecules, particularly proteins. Recently, a mechanism underlying phosphate’s unique properties was decoded. Unlike the SCN^−^ anion, which increases the μs-ms backbone dynamics of proteins by disrupting hydrophobic interactions and destabilizing their thermodynamic stability [124], phosphate significantly enhances backbone μs-ms dynamics without disrupting hydrophobic interactions, instead strengthening thermodynamic stability [125]. This ability is likely due to the exchange between different ionic states of phosphate within the hydration shell of proteins, a process that also occurs on the μs-ms time scale [134] (Figure 4C). This mechanism may explain the unique ability of phosphate-containing molecules, such as ATP, ADP, diphosphates, triphosphates, and polyphosphates, to promote protein folding and enhance protein stability. These molecules likely achieve this by facilitating transitions among different protein conformational states. Indeed, ribosomes were initially proposed to act as chaperones for the folding of nascent polypeptides [263,264,265,266]. However, it was later discovered that it is actually the rRNA molecules within the ribosome that enhance this folding process. It is now established that nucleic acids, in general, can function as chaperones to assist in protein folding [267,268,269].

This unique ability of polyphosphates, combined with their capacity for high-energy storage, may serve as a key foundation for their selection as initial scaffolds for LLPS of primitive polypeptides, thus driving the formation of protocells (Figure 7C). Intriguingly, triphosphate exhibits the highest capacity for inducing protein folding. Surprisingly, while phosphate and diphosphate demonstrate weaker abilities, tetraphosphate does not show any improved folding capacity. For instance, adenosine 5′-(pentahydrogen tetraphosphate) (ATPP) has the same capacity to induce folding as ATP but additionally acquires the ability to trigger aggregation. Furthermore, the inducing capacity is also dependent of the atoms that link the phosphates. Replacing the oxygen atom connecting the beta and gamma phosphates of ATP with carbon or nitrogen not only reduces its ability to induce folding to a level comparable to that of ADP but also introduces the ability to promote aggregation, a trait that is absent in both ATP and ADP [128].

In this context, triphosphate appears to be maximized not only for an immediate reactant in probiotic evolution and energy storage [259,260,261,262] but also for enhancing protein stability and folding in an energy-independent manner, utilizing its unique feature associated with hydration (Figure 8A). However, triphosphate, diphosphate, and polyphosphates are all inorganic molecules, which come with several drawbacks: (1) they strongly trigger the aggregation of partially folded and disordered proteins with significant exposure of hydrophobic patches through their strong electrostatic screening effects [128]. As a consequence, they primarily function as chaperones in single-celled organisms, where the proteomes are predominantly composed of well-folded proteins. (2) Their interactions mostly with polar or charged surface residues of proteins are relatively weak, resulting in limited effectiveness in enhancing the stability and folding even for folded proteins [125] and in modulating phase separation of IDRs [73]. (3) They lack the ability to interact with hydrophobic surfaces, which are characteristic of biomacromolecules, and cannot effectively modulate hydrophobic interactions, a key driving force in protein folding.

Marvelously, by linking inorganic, charged triphosphate to organic, hydrophobic adenosine, nature has created ATP, which successfully bridges the inorganic and organic worlds existing in water (Figure 8). The unique structure of ATP enables it to interact broadly with hydrophobic patches on various biomacromolecules, particularly proteins, enhancing its affinity through multiple-site interactions. Notably, the aggregation-inducing properties of triphosphate and diphosphate appear to be mitigated by the covalently linked adenosine; ATP and ADP no longer trigger the aggregation of disease-causing mutants, such as C71G-PFN1, which exhibit deficient tertiary packing. Instead, ATP and ADP increase the stability of such proteins. These results suggest that in ATP and ADP, adenosine interacts with triphosphate, preserving the hydration-mediating capacity of triphosphate while simultaneously shielding its aggregation-inducing properties. Importantly, ATP, which closely resembles nucleotides, the building blocks of nucleic acids, has the ability to compete with nucleic acids for interactions with proteins and can even directly influence their conformations [270,271]. As a result, ATP not only serves as the universal energy currency for all living cells but also possesses the capacity to modulate protein homeostasis in an energy-independent manner [19]. Additionally, ATP plays a role in shaping the interfaces between the genome and proteome [139].

As illustrated in Figure 8A, ATP, proteins, and nucleic acids can all interact with one another in an energy-independent manner. Nucleic acids can specifically bind to the DNA/RNA-binding pockets of proteins, while also exerting non-specific electrostatic effects. They interact with Arg residues in IDRs through π-π and π-cation interactions, and with Lys residues in IDRs via π-cation interactions to promote LLPS. Interestingly, nucleic acids can enhance protein folding and stability by modulating hydration. Similarly, ATP can exert comparable effects on proteins but with different specificities and affinities. Additionally, ATP imposes hydrotropic effects, cosolute effects, and screening effects. It also influences nucleic acids through non-specific electrostatic interactions, hydrotropic effects, cosolute effects, and screening effects. Through the complex interplay of these interactions, ATP acquires the energy-independent ability to effectively induce protein folding, inhibit aggregation, and enhance thermodynamic stability.

Consequently, in modern cells, ATP not only serves as the universal energy currency driving all biological processes but also functions as a molecular chaperone in an energy-independent manner [123]. It acts as a structural stabilizer [128,143,144,228,272,273,274,275], a modulator of LLPS [73,136,137,138,142,190,225,226,227,228,229,230,231,274,275,276,277,278,279,280,281,282,283,284,285], an aggregation inhibitor [123,143,286,287,288,289,290,291], and a crowding antagonist [145,146,147] (Figure 8B). In contrast, nucleic acids, with their multiple binding sites compared to ATP, can instigate LLPS in IDRs that contain numerous Arg/Lys residues [190,225]. Notably, although nucleic acids can achieve a high binding affinity to IDRs through multivalent but independent interactions with multiple Arg/Lys residues, this binding is relatively susceptible to displacement by ATP. This susceptibility arises because these dynamic complexes are highly accessible, allowing ATP to simultaneously displace each independent binding event involving nucleic acids. ATP’s triphosphate group also possesses a unique ability to interact with water molecules and the side chains of Arg/Lys residues. These distinctions may influence the competition between ATP and nucleic acids for binding to Arg/Lys residues and their respective abilities to modulate LLPS. This phenomenon may explain why a substantial proportion of IDRs contain multiple Arg/Lys residues, making them prone to phase separation upon induction by nucleic acids of typical lengths. However, the occurrence of MLOs is notably limited within cells. It is likely that the high cellular concentrations of ATP inhibit or dissolve most nucleic-acid-induced phase separations of IDRs. Consequently, proteins within MLOs tend to feature folded domains with high nucleic-acid-binding affinities, enabling them to establish interactions with nucleic acids even in the presence of elevated ATP concentrations [139,285].

## 10. Summary and Challenges

All known life forms, including viruses, develop from genetic information compressed in a one-dimensional sequence of nucleic acids, either DNA or RNA (I of Figure 9A) [10,47,48]. The four-nucleotide coding system of nucleic acid molecules offers several advantages over the binary coding (0 and 1) used in computing [292]. These include (1) a higher information density, allowing more data to be stored per unit; (2) built-in error tolerance through redundancy and repair mechanisms; (3) biological efficiency, enabling evolutionary flexibility; and (4) the ability to encode not only protein structures but also regulatory sequences, epigenetic markers, and other complex information. Decoding the compressed genetic information to form a functional cell, and ultimately a fully developed organism, requires multiple complex steps. Among these, protein synthesis is particularly critical, as proteins serve as the primary functional molecules and structural components that drive nearly all cellular processes. Proteins are composed of 20 naturally occurring amino acids, which can be categorized into a simplified five-letter amino acid system (II of Figure 9A) [293]. Proteins spontaneously fold into an extremely diverse spectrum of conformations, including folded domains, IDRs, and membrane proteins, which can further assemble into MLOs or integrate into membrane-enveloped organelles such as mitochondria, the endoplasmic reticulum, and the cell membrane.

Protein folding, the process by which linear polypeptides spontaneously self-assemble into functional three-dimensional structures, is recognized as one of biology’s greatest mysteries. In his Nobel Lecture, Anfinsen stated that “the native conformation is determined by the totality of interatomic interactions and hence by the amino acid sequence, in a given environment” [12]. However, predicting protein structures from first principles still remains a significant challenge. Nonetheless, AI systems like AlphaFold 2 and 3 [294,295,296], built using artificial neural network architectures [297,298,299], have made an unprecedented breakthrough by deep-learning existing protein structures and subsequently achieving accurate structure predictions.

The failure to predict protein structures from first principles has long been attributed to our limited understanding and ability to computationally model protein hydration [300,301,302,303,304,305]. Despite this, AlphaFold 2 and 3, trained on existing protein structures without hydration information, have accurately predicted protein structures and complexes, also bypassing the need for detailed knowledge of microscopic mechanisms. This success implies that not only the three-dimensional structures, as noted by Anfinsen [12], but also hydration information might be indirectly coded in the protein sequence. In this context, hydration structures and dynamics are also determined by the properties of proteins specified by the sequence. For instance, the cataract-causing G18V mutation in γS-crystallin disrupts its hydration shell, preventing ATP from antagonizing the crowding-induced destabilization of the mutant. In contrast, other cataract mutations like D26G, S39C, and V42M only partially damage the hydration shell [146,147,148].

All known life forms exist in water, and proteins are also highly hydrated. The unique findings reviewed here, including the effects of salts on protein solubility, membrane interactions, stability, and dynamics, as well as ATP’s role in modulating crowding-induced destabilization, protein folding, and phase separation of IDRs, all highlight the central role of protein hydration in these processes. Although the interactions driving protein folding are encoded in the amino acid sequence, it is water that allows the realization of these interactions such as the hydrophobic effect, hydrogen bonding, electrostatic screening, and solvent effects. Water, in this context, acts as the medium through which the folding potential is ‘decoded’ and expressed. Without water, the forces required for proper protein folding would not manifest correctly, leading to misfolded or non-functional proteins.

Therefore, protein hydration has at least three fundamental roles: (1) It ensures that proteins, regardless of their properties, remain soluble in unsalted water (III of Figure 9A), the medium in which proteins might have originally evolved. (2) Protein–water interactions shape the energy landscape of protein folding. The release of bound water molecules into the bulk solvent entropically drives protein folding, together with the specific interactions encoded in the protein sequence and influenced by environmental factors. This release of water molecules facilitates proteins to overcome energy barriers and to be “self-decoded” into various cellular states, including folded (IV of Figure 9A), intrinsically disordered (V of Figure 9A), phase-separated (VI of Figure 9A), membrane-associated (VII of Figure 9A), and aggregated or amyloid (VIII of Figure 9A) forms. (3) Each of these cellular states of proteins is also uniquely hydrated, with dynamics occurring on different time scales that are essential for their functionality. These dynamics are modulated by various molecules, including universally by salt ions and ATP.

These findings have profound and far-reaching implications, not only for advancing our fundamental understanding of the principles underlying life’s molecular architecture and functions but also for driving practical applications across diverse fields such as drug design, biotechnology, and synthetic biology. Additionally, they offer new perspectives on deciphering the mechanisms of protein-aggregation diseases and developing effective therapeutic strategies. For instance, water, ATP, and ions are ubiquitous and indispensable components of all living systems, yet their previously underappreciated contributions may help explain challenges faced in designing drugs with high affinity and specificity or in engineering proteins with desired properties based on first principles. The neglect of these factors in traditional approaches might have led to oversights, particularly in addressing dynamics of molecular interactions and environmental conditions that in fact significantly influence biological processes. Furthermore, disease-causing mutations in the IDRs of proteins such as ALS-causing TDP-43 and FUS might have profoundly affected hydration dynamics, interactions with salts, and ATP binding, potentially altering protein behavior. These changes may cascade to affect the structural organization and functional topologies of associated cellular networks, exacerbating pathological outcomes. Addressing these complexities is expected to pave the way for innovative therapeutic interventions that specifically target these previously unrecognized molecular mechanisms.

The analysis presented here has also highlighted three fundamental challenges that requires future exploration:

1. How does the protein sequence specify hydration, and how is protein hydration modulated, particularly by ATP? Emerging results suggest that the formation of protein hydration by its sequence is a highly specific process. For instance, while ATP is able to effectively antagonize crowding-induced destabilization of the wild-type γS-crystallin even at a 1:1 ratio by mediating its hydration shell [145,146,147], the cataract-associated G18V mutation is sufficient to dramatically disrupt its hydration shell [165], rendering ATP ineffective in countering the crowding-induced destabilization of the mutant [146]. Intriguingly, other cataract mutations, such as D26G, S39C, and V42M, only partially compromise the hydration shell and remain partially responsive to ATP modulation [147]. So, how can a single mutation, such as G18V, completely disrupt the hydration shell? How is ATP modulation achieved? By altering the shape, thickness, or dynamics of the hydration shell?

In the future, the development and application of emerging experimental and computational techniques will be crucial for overcoming current limitations in understanding protein hydration dynamics. These advancements may include cutting-edge cryo-electron microscopy (cryo-EM) for capturing high-resolution structural insights, artificial intelligence (AI)-based simulations for predicting dynamic interactions, and deep learning tools for molecular modeling that integrate vast amounts of experimental and theoretical data. Such approaches could enable unprecedented precision in mapping hydration networks, understanding their role in protein folding and function, and deciphering complex biomolecular interactions. By bridging gaps in our knowledge, these innovations have the potential to transform how we study and manipulate proteins in biological and therapeutic contexts.

2. In addition to ATP, various phosphate-containing biomolecules exist in cells, including GTP, CTP, TTP, UTP, FAD, NAD/NADP, CoA, and Poly(ADP-ribose) (PAR). Their hydration dynamics and roles in modulating protein hydration remain to be explored, despite their relatively low concentrations in cells. Intriguingly, hydroxyapatite is not only a key structural component for mineralization, forming tusks, horns, bones, and teeth in many organisms, but also plays vital functional roles, such as enabling balance and motion sensing in fish otoliths, crucial for underwater orientation, and serving as a reservoir for calcium and phosphate ions to regulate physiological processes. It is of fundamental interest to assess whether the unique hydration and dynamic behaviors of phosphate contribute to its functions. In particular, two key biomacromolecules, nucleic acids and phospholipids, also contain phosphate groups (Figure 9B). Although most phosphate groups in nucleic acids and phospholipids exhibit only two ionic states (Figure 9B), unlike the more varied ionic states of triphosphate or polyphosphate groups, they still exchange on a relatively slower time scale (μs-ms) across a wide pH range, especially in physiologically relevant conditions [306]. Given their ubiquitous presence in all cells at high densities, the effects of phosphate groups on the dynamics of nucleic acids, phospholipids, and other biomolecules cannot be overlooked. Indeed, nucleic acids have already been shown to act as chaperones in protein folding [267,268,269]. In contrast, the role of phosphate groups in the dynamics of phospholipids remains almost unknown. How do these groups contribute to the dynamics and functions of phospholipids? Do phospholipid surfaces serve as hubs that influence protein folding and aggregation? If so, what are the underlying mechanisms? Indeed, disease-associated protein mutants have been extensively observed to aggregate on the surfaces of phospholipid membranes [39].


3. The most fundamental mystery in biology is how cellular networks operating at the molecular levels leads to the emergence of functionality and morphogenesis in cells, and ultimately, in whole organisms. It is well-established that cells contain highly complex networks, primarily metabolic, gene regulatory, and protein–protein interaction (PPI) networks [307,308], which involve numerous molecules and environmental factors (Figure 9C). These networks are essential for determining how cells differentiate and develop into tissues and organs by interpreting the genetic blueprint encoded in nucleic acids to build the organism’s physical structure. It was previously hypothesized that biological functions and morphogenesis arise as an ‘emergent’ phenomenon from these complex networks because networks are universal approximators for almost any functions [309,310,311,312,313]. However, despite extensive research across multiple disciplines, the exact mechanisms cannot be fully understood or reconstructed by analyzing the individual components of these systems or network nodes alone, primarily due to their non-linear nature [314].

While it is well-established that the DNA within an egg encodes all the genetic information necessary to build an organism, recent findings emphasize the crucial role of pre-existing cellular networks, including specific cytoplasmic factors, in decoding the information. This situation can be compared to artificial neural networks in AI, which rely on pre-trained architectures to process data. In a similar manner, pre-existing cellular networks help interpret the developmental information encoded in DNA. Thus, both DNA sequences and the pre-existing cellular networks are required for organismal development. This analogy raises the intriguing possibility that biological information is also stored in a distributed fashion across cellular networks, much like weights and biases are distributed across nodes in generative AI models. If this hypothesis holds, it has several fundamental implications: (1) Data derived from experiments using biochemical, biophysical, and cellular and molecular biology methods might only represent fragments of large cellular networks as well as the weights and biases stored in their nodes. Such data could be unreadable or uninterpretable without the full network context. (2) Proper organismal development depends not only on the genetic information encoded in DNA but also on the pre-existing topological architecture of cellular networks. Both genetic sequences and cellular networks are essential and may be inheritable, implying a more complex model of inheritance than currently thought. (3) Even transient structures, such as water hydrogen-bonding networks and hydration dynamics, occurring on timescales from picoseconds to nanoseconds, might be encoded in cellular networks by modifying the weights or the biases of their nodes, or even reshaping by their topologies. For example, the vast number of phase-separated states of IDRs have been proposed as a mechanism for storing “cellular memory” [15]. Moreover, mutations in IDRs, as seen in prion-like diseases involving proteins such as TDP-43 and FUS, may disrupt protein hydration, leading to alterations in the parameters or/and even architectures of cellular networks.

Cellular networks are far more complex than the artificial neural networks used in current AI systems. First, while AI systems rely on rigid silicon chips with limited flexibility, organisms utilize a diverse range of biomacromolecules and small molecules that exhibit flexibility across various timescales, forming dynamic networks that enhance resilience and adaptability. Second, AI components are rigidly aligned in a vacuum, whereas cellular networks operate in highly hydrated environments, suspended in water. Third, the topological architectures of AI networks are fixed, but cellular networks continuously reorganize and evolve in response to the cell’s needs. Fourth, AI networks are uniformly powered by electricity, while cellular networks rely on spacetime-specific energy sources, primarily the chemical energy from ATP. This enables both energy efficiency and precise, localized control over network activity, allowing different regions of the cell to be activated or deactivated as needed. Lastly, multiple distinct networks with unique architectures operate non-uniformly yet cooperatively within a single cell and likely across entire organisms, adding another layer of complexity to biological functionality.

It is increasingly evident that even the artificial neuron, the basic computational unit of artificial neural networks, is not a natural unit for human understanding. This is because many artificial neurons are polysemantic, responding to combinations of seemingly unrelated inputs [315]. The profound complexity of cellular systems, characterized by dynamic architectures, hydration dependence, and flexible interaction architectures, indicates that biological processes can achieve sophistication far surpassing current AI. One intriguing hypothesis is that quantum mechanics may underpin certain biological processes, enabling quantum operations. While the exact mechanisms remain unknown, such explorations could transform our understanding of biology and drive breakthroughs in cognitive science and AI [316]. For example, consciousness has been proposed to emerge from quantum computations within neurons [316,317,318].

## Figures and Tables

**Figure 1 ijms-25-12817-f001:**
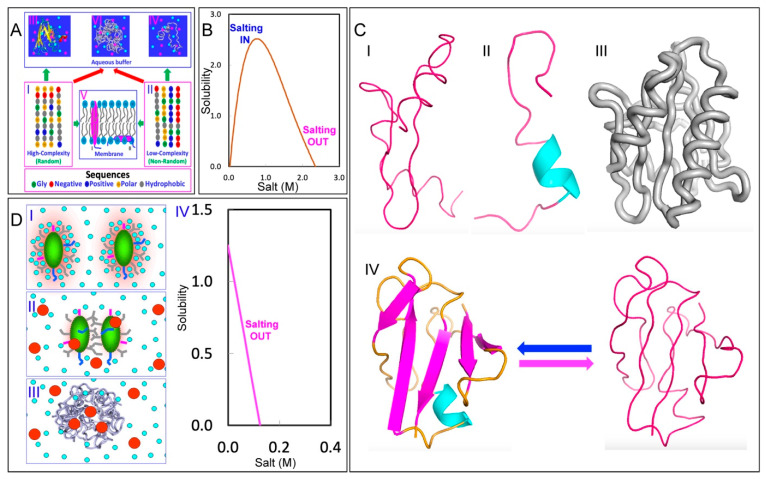
The effect of salts on protein solubility beyond “salting-in/salting-out”. (**A**) Sequence–structure relationship of proteins. Based on the sequences as represented by five types of amino acids, proteins can be classified into high complexity or random (I), and low-complexity or non-random (II) sequences. A portion of proteins of the high-complexity sequence can fold into uniquely folded structures soluble with high concentrations of salts (III), while many of proteins of the low-complexity sequence remain intrinsically disordered (IV). Furthermore, ~30% proteins of both high- and low-complexity sequences can fold in the membrane environments (V). Interestingly, a large number of proteins of both high- and low complexity sequences appear to be insoluble in vivo with high concentrations of salts (VI). (**B**) The classic curve of protein solubility versus salt concentration for a well-folded protein which consists of “salting-in” and “salting-out” phases. (**C**) Solution conformations of insoluble proteins solubilized in unsalted water. Four groups of conformations have been observed so far: (I) highly disordered state without any stable secondary and tertiary structures; (II) partially folded state with some secondary but without tertiary structures; (III) molten globule-like state with secondary structures as well as dynamic tertiary packing; and (IV) coexisting folded and unfolded states exchanging in equilibrium. (**D**) A model to rationalize how salt ions affect the solubility of an insoluble protein. (I) An unrefoldable and insoluble protein in unsalted water with the solution pH several units away from its pI. Small cyan spheres stand for water molecules and green ellipsoids for protein molecules with a large number of hydrophobic side chains exposed. (II) Protein molecules in the presence of a small number of salt ions (larger red spheres). In addition to imposing non-specific electrostatic screening, the presence of salts also provides specific anion binding to protein residues, thus altering the surface electrostatic potential. Furthermore, salts also change the hydration structure as well as protein dynamics, as represented by the broken lines of green ellipsoids. (III) The complex interplay of these salt effects results in aggregation of the protein. (IV) The unique curve of protein solubility versus salt concentration for an insoluble protein. Figure 1A was adapted from Figure 1a in Ref. [19].

**Figure 2 ijms-25-12817-f002:**
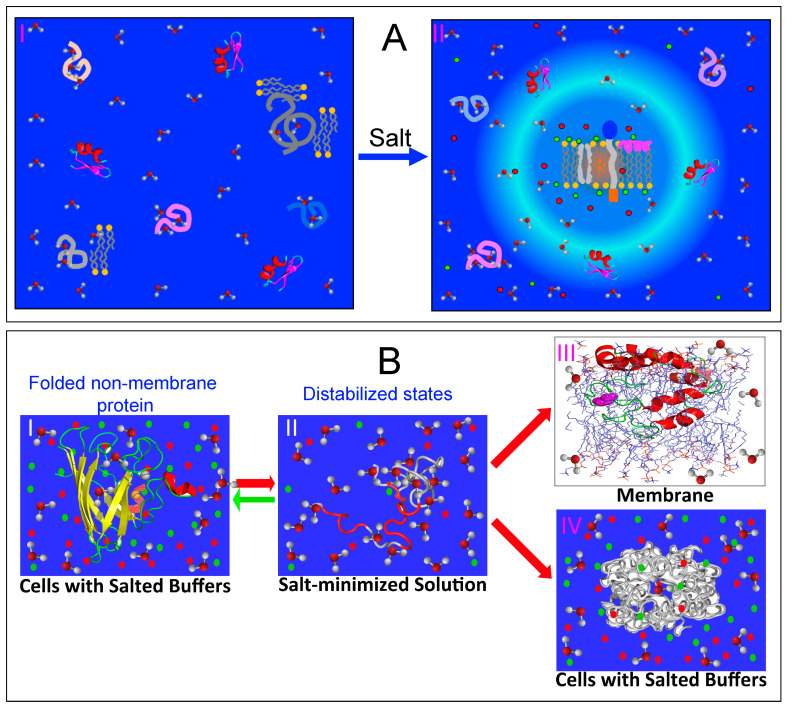
Emergence of membrane proteins and transformation of a folded cytosolic protein into a membrane-interacting protein. (**A**) (I) In the primitive water body with minimized salt concentrations, all proteins regardless of sequences are soluble. (II) The increase in the salt concentrations drives highly hydrophobic proteins to form membrane proteins. (**B**) Via disease-associated factors, a folded cytosolic protein (I) can be disrupted into a highly disordered form, which is only soluble in salt-minimized water (II). As this highly disordered form universally contains hydrophobic and/or amphiphilic patches, it acquires the novel capacity to interact with membranes energetically, driven by folding into non-native helix/loop structures in membranes (III), or becomes aggregated in the presence of high salt concentrations (IV).

**Figure 4 ijms-25-12817-f004:**
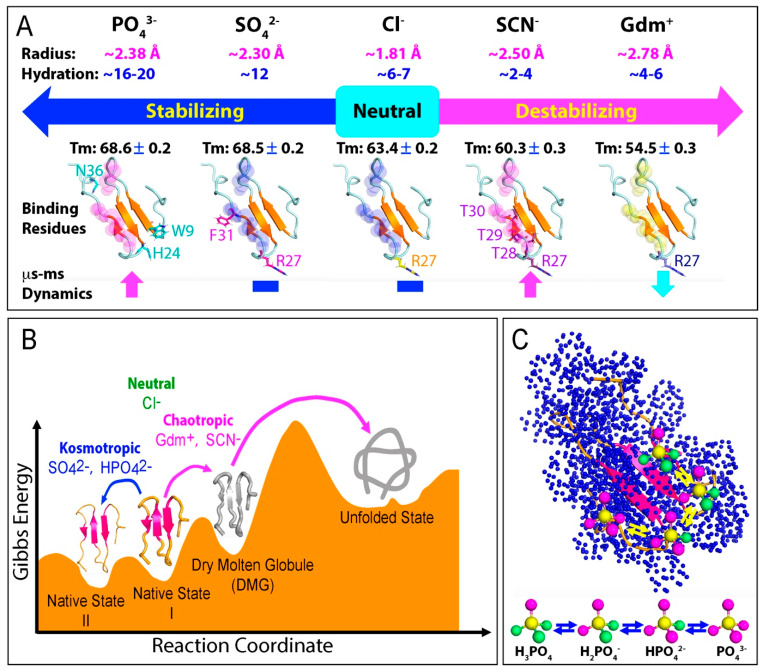
**Proposed mechanisms for the Hofmeister effects.** (**A**) The effects of the four anions and the Gdm+ cation on the conformation, thermodynamic stability, binding residues, and backbone dynamics of the WW4 domain. The ions are ranked based on the Hofmeister series, with the ionic radius and water molecule numbers indicated. The thermodynamic stability is reported by the melting temperature (Tm) values of WW4 in the absence and in the presence of, respectively, the five salts at 200 mM. In the structure of WW4, the spheres are utilized to indicate six residues: Gly17, Val18, Arg27, Thr28, Thr29, and Thr30, with significant μs-ms backbone dynamics (ΔR^2^_eff_ > 4 Hz at the 500 MHz field), while the sticks are used to show the residues with significant binding as defined by the chemical shift difference (CSD) in the presence of, respectively, the five salts at 200 mM. The changes in μs-ms backbone dynamics upon adding, respectively, the five salts at 200 mM are also indicated. (**B**) A diagram illustrating the effects of five salts on the folding and stability of the WW4 domain. (**C**) The proposed mechanism for the phosphate anion to provoke μs-ms dynamics of the WW4 domain by exchanges of the ionic states on a μs-ms time scale. Figure 4A was adapted from Figure 7 in Ref. [125].

**Figure 5 ijms-25-12817-f005:**
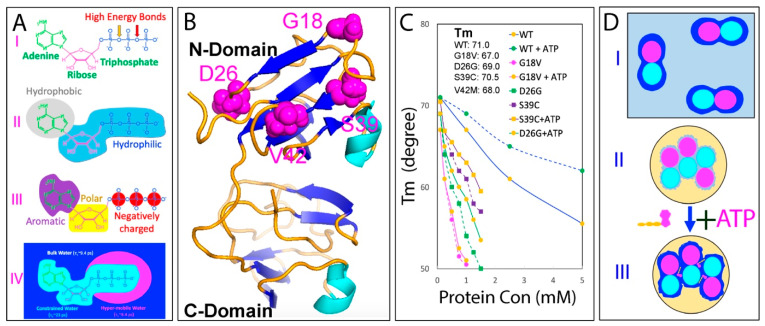
**ATP antagonizes the crowding-induced destabilization of γS-crystallin.** (**A**) ATP has unique structural properties and thus may act as (I) energy currency via the hydrolysis of high-energy bonds; (II) a biological hydrotrope with the presence of hydrophobic adenine and hydrophilic ribose and triphosphate; (III) a bivalent binder via the aromatic purine ring and highly negatively charged triphosphate chain; and (IV) a hydration mediator, resulting from its unique hydration structure, previously derived from the results of microwave dielectric spectroscopy, which was modeled to contain “constrained water” with a dielectric relaxation time (τc) of ~23 ps, as well as “hyper-mobile water” with a τc of ~8.4 ps, even smaller than that of bulk water (9.4 ps). (**B**) Three-dimensional structure of human γS-crystallin with four cataract-causing mutation residues displayed in spheres and labeled. (**C**) Concentration-dependence curves of melting temperature (Tm) of WT, G18V, D26G, and S39C γS-crystallins without and with ATP at 20 mM. (**D**) A speculative model to illustrate that ATP antagonizes the crowding-induced destabilization by targeting protein hydration. Under the extremely crowded condition, the hydration shell (cyan) is disrupted/twisted. However, with the presence of ATP, the disrupted/twisted hydration shell (cyan) is restored to some extent. Figure 5A was adapted from Figure 1b in Ref. [19].

**Figure 6 ijms-25-12817-f006:**
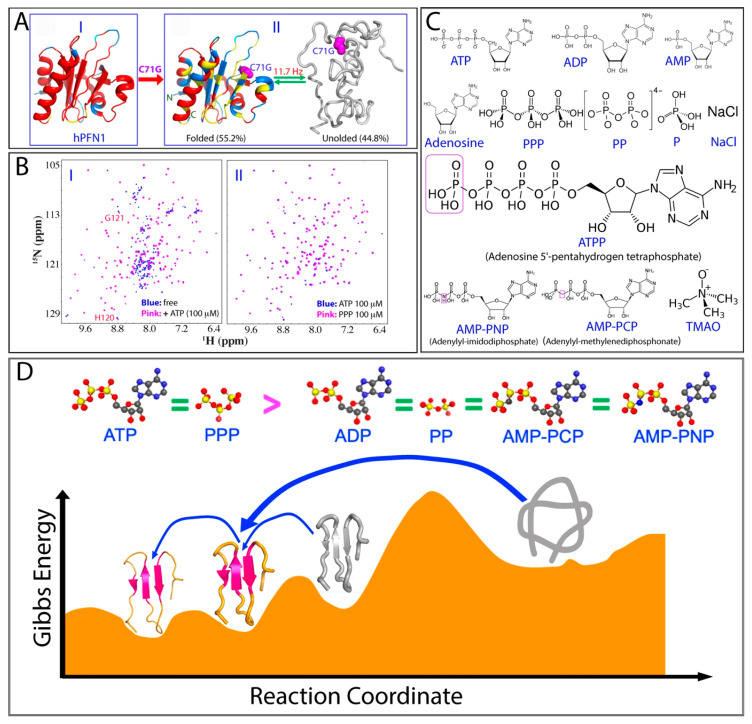
**ATP induces protein folding at the highest efficiency.** (**A**) A diagram to show the ALS-causing C71G mutation destabilizing hPFN1 (I) into the coexistence of the folded (with the average S^2^ value of 0.73) and unfolded states (II). The populations of the folded and unfolded states have been calculated to be, respectively, 55.2% and 44.8%, which undergo a conformational exchange at 11.7 Hz. (**B**) (I) Superimposition of HSQC spectra of ^15^N-labeled C71G-hPFN1 at a concentration of 50 μM in the absence (blue) or in the presence of ATP (pink) at 100 μM. (II) Superimposition of HSQC spectra of C71G-hPFN1 in the presence of ATP at 1:2 (blue) and triphosphate (PPP) (pink) at 1:2. (**C**) Chemical structures of ATP and 11 related compounds. (**D**) Ranking of the capacity of ATP-related compounds in enhancing the conversion of the unfolded state into the folded state of ALS-causing C71G-hPFN1. Figure 6A was adapted from Figure 2a in Ref. [128], while Figure 6C was adapted from Figure 1a in Ref. [128].

**Figure 7 ijms-25-12817-f007:**
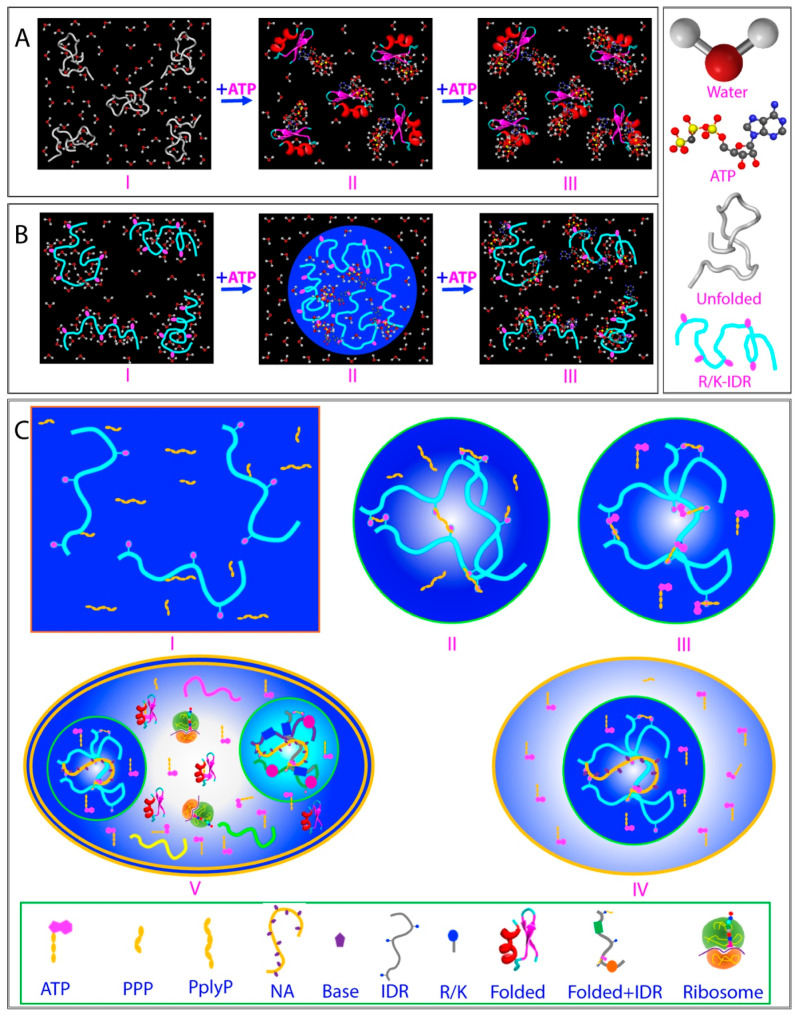
**ATP induces folding, modulates LLPS, and drives the emergence of protocells.** (**A**) ATP induces protein folding entropically, driven by the release of water molecules bound to the unfolded state into bulk solvent. (**B**) ATP induces LLPS of Arg-/Lys-containing IDRs entropically driven by the release of water molecules bound to the unfolded state into bulk solvent, followed by dissolution of LLPS with the excessing binding of ATP. (**C**) The emergence of protocells driven by polyphosphates, ATP, and nucleic acids.

**Figure 8 ijms-25-12817-f008:**
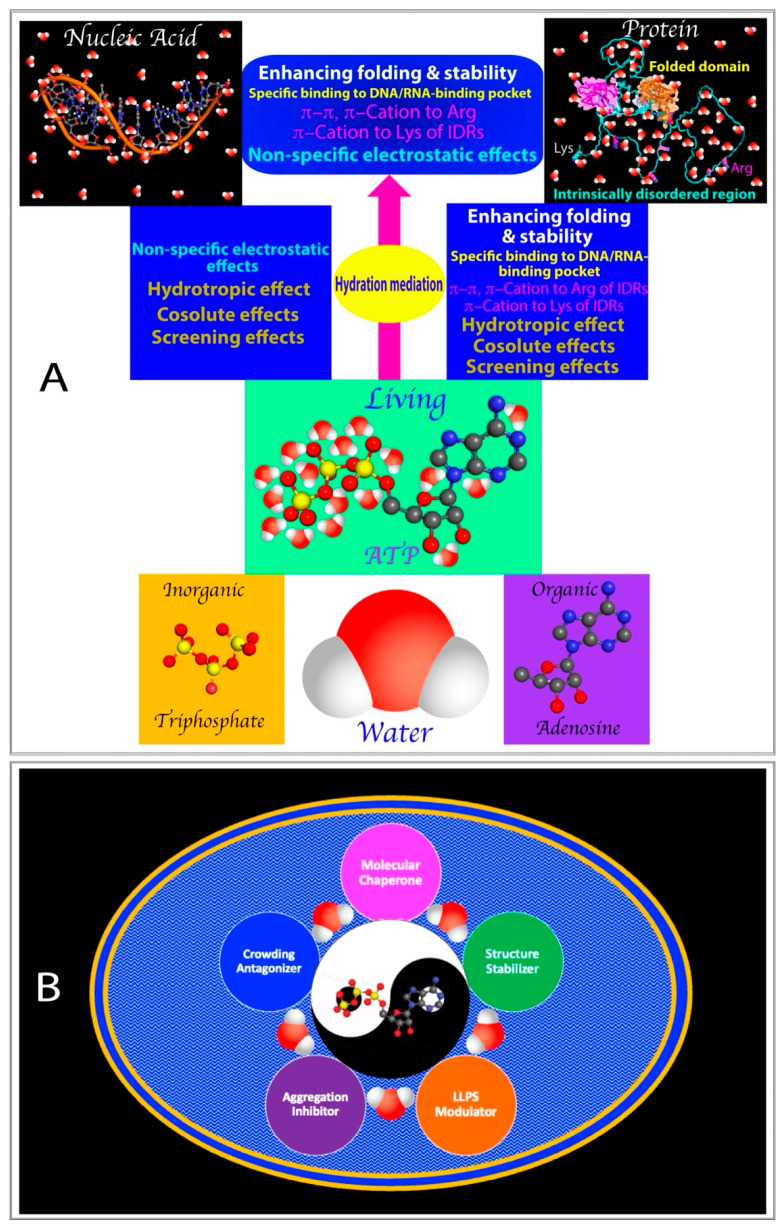
**The emergence and energy-independent functions of ATP.** (**A**) By linking organic, charged triphosphate and organic, hydrophobic adenosine, ATP acquires the capacity to site-specifically and non-specifically interact with proteins and nucleic acids. With a high similarity to nucleic acids, ATP thus has the capacity to shape the interface between the genome and the proteome through the evolutionary trajectory. (**B**) Energy-independent functions of ATP on protein homeostasis in modern cells.

**Figure 9 ijms-25-12817-f009:**
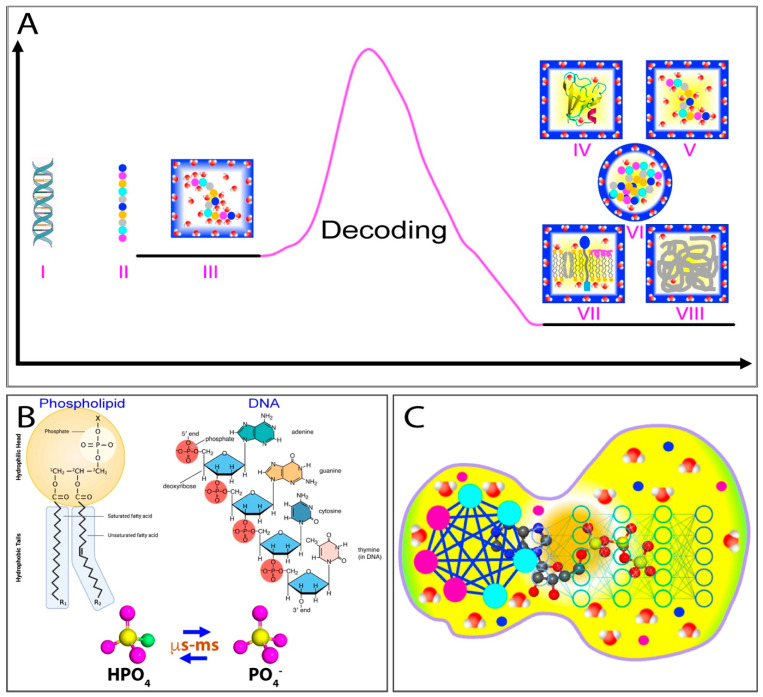
**Water “self-decodes” the protein sequence and cellular networks in water.** (**A**) Decoding of genetic information compressed in DNA (I) into the protein sequence (II). In pure water, all proteins are highly hydrated and soluble (III), which could be driven by the release of bound water molecules as well as the interplay of intrinsically encoded interactions and environmental factors to cross the energy barrier to form folded (IV), intrinsically disordered (V), phase-separated (VI), membrane-associated (VII), and aggregated (VIII) states. (**B**) The chemical structures of the phospholipid and DNA, to illustrate the presence of a phosphate group that might provoke μs-ms dynamics. (**C**) A cell constituted by the extremely complex networks, which are operating cooperatively in membrane-compartmentalized aqueous spaces rich in salts and spacetime-specifically energized by ATP.

## Data Availability

No new data were created or analyzed in this study.

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
