# Peer review of "In the Beginning: Let Hydration Be Coded in Proteins for Manifestation and Modulation by Salts and Adenosine Triphosphate"

_ijms, 2024, doi:10.3390/ijms252312817_

Round 1

Reviewer 1 Report

Comments and Suggestions for Authors

The review article reads fine currently but addressing a few things as noted below will provide a more comprehensive review.

The authors can expand on how can the role of hydration and ATP in protein dynamics be applied to fields such as drug design, biotechnology, or synthetic biology. How these insights could be used to tackle diseases like Alzheimer's or ALS could also be included.

While the authors have discusses ATP as a modulator of protein folding and phase separation, more specific mechanistic details or hypotheses on how ATP’s effects differ across protein types (e.g., membrane proteins, cytosolic proteins) would enhance the depth of the review.

The review touches on protein aggregation in neurodegenerative diseases but could expand on how water and ATP directly influence these pathological processes. Including specific examples from diseases like Alzheimer's or Parkinson's with molecular models could strengthen the discussion.

The review should focus more on the emerging techniques. Currently, it lacks a detailed discussion on emerging experimental or computational techniques that could address the current limitations in understanding protein hydration dynamics (e.g., advanced cryo-EM, AI-based simulations, or deep learning tools for molecular modeling).

The review highlights phosphate’s influence on protein dynamics but could further explore its implications beyond hydration shells, such as its involvement in broader cellular processes like signaling, energy metabolism, or membrane dynamics.

Author Response

Point-to-point Response to Reviewer 1

First I would like to express my sincere gratitude to the reviewer for the critical comments which serve to improve the revised manuscript significantly. Please kindly find below the point-to-point response and all the revised parts are colored in blue in the revised manuscript.

  1. The authors can expand on how can the role of hydration and ATP in protein dynamics be applied to fields such as drug design, biotechnology, or synthetic biology. How these insights could be used to tackle diseases like Alzheimer's or ALS could also be included.

While the authors have discusses ATP as a modulator of protein folding and phase separation, more specific mechanistic details or hypotheses on how ATP’s effects differ across protein types (e.g., membrane proteins, cytosolic proteins) would enhance the depth of the review.

The review touches on protein aggregation in neurodegenerative diseases but could expand on how water and ATP directly influence these pathological processes. Including specific examples from diseases like Alzheimer's or Parkinson's with molecular models could strengthen the discussion.

Response: thanks so much for the critical comments/suggestions. I have developed a paragraph in the last section to discuss these points.

  1. The review should focus more on the emerging techniques. Currently, it lacks a detailed

discussion on emerging experimental or computational techniques that could address the current limitations in understanding protein hydration dynamics (e.g., advanced cryo-EM, AI-based simulations, or deep learning tools for molecular modeling).

Response: thanks so much and I have added a paragraph to discuss this point in the last section.

  1. The review highlights phosphate’s influence on protein dynamics but could further explore its implications beyond hydration shells, such as its involvement in broader cellular processes like signaling, energy metabolism, or membrane dynamics.

Response: thanks so much and I have added two short paragraphs into the revised manuscript.

Reviewer 2 Report

Comments and Suggestions for Authors

The manuscript titled “In the Beginning: Let Hydration be Coded in Protein for Manifestation and Modulation by Salts and ATP” by Song, J. is a Review work where the author outlined thr most recent advances in molecular dynamics to resolve the protein structure when the proteins are interaction in water solutions. The role of energetic molecules and their impact on protein structures are also addressed. The manuscript is generally well-written. However, it exists some points that need to be addressed (please, see them below detailed point-by-point) to improve the scientific quality of the submitted manuscript paper before this article will be consider for its publication in the International Journal of Molecular Sciences.

1) The author should consider to add the term “intrinsically disordered regions” in the keyword list.

2) “Water (…) formed in the early universe (...) water molecules in interstellar space” (lines 38-41). According to this statement, why did not exist water in many planets? Some explanation should be furnished in this regard.

3) Then, in the introduction section the author should state the pivotal role of protein functions in life providing some quantitative data details. This will significantly aid the potential readers to better understand the significance of this Review work.

4) “In this regard (…) NMR spectroscopy (…)” (line 114). “Marvelously, (…) nuclear magnetic resonance (NMR) investigations” (lines 219-220). The abbreviation should be defined the first time that the full-name appears in the text. This comment should be taken into account for the rest of the main manuscript sections.

5) “Protein solubility is essential (…) protein aggregation and amyloidogenesis (…) like Alzheimer’s disease (AD), Parkinsons’s disease (PD) (…) and frontotemporal dementia (FTD)” (lines 169-175). Here, even if I agree with the information provided by the author it should be also remarkable to discuss how other key factors as the presence of certain divalent cations [1] or the ionic strength [2] can also lead to the formation of amyloidogenic fibrils. This will strengthen the multiparametric factors that can trigger the onset of neurodegenerative malignancies and the importance to study them.

[1] Carapeto, A.P.; et al. Morphological and Biophysical Study of S100A9 Protein Fibrils by Atomic Force Microscopy Imaging and Nanomechanical Analysis. Biomolecules 2024, 14, 1091. https://doi.org/10.3390/biom14091091

[2] Ziaunys, M.; et al. Polymorphism of Alpha-Synuclein Amyloid Fibrils Depends on Ionic Strength and Protein Concentration. Int. J. Mol. Sci. 2021, 22, 12382. https://doi.org/10.3390/ijms222212382

6) “3. Transformation of the folded cytosolic proteins into membrane-interacting proteins” (lines 328-393). A schematic representation will benefit the potential readers to have a more complete overview of the crosstalk among these two protein states.

7) “10. Summary and challenges (lines 1207-1372). This section perfectly remarks the most relevant outcomes found by the author in this field and the promising future perspectives and some limitations to be overcome.  It should be desirable to add a brief statement to discuss about the potential future action lines to pursue the topic covered in this research.

Author Response

Point-to-point Response to Reviewer 2

First I would like to express my sincere gratitude to the reviewer for the critical comments which serve to improve the revised manuscript significantly. Please kindly find below the point-to-point response and all the revised parts are colored in blue in the revised manuscript.

1) The author should consider to add the term “intrinsically disordered regions” in the keyword list.

Response: many thanks and I have added it.

2) “Water (…) formed in the early universe (...) water molecules in interstellar space” (lines 38-41). According to this statement, why did not exist water in many planets? Some explanation should be furnished in this regard.

Response: thanks so much for the critical comment. I have added sentences to clarify it.

3) Then, in the introduction section the author should state the pivotal role of protein functions in life providing some quantitative data details. This will significantly aid the potential readers to better understand the significance of this Review work.

Response: thanks so much for the important suggestion. I have added a paragraph into the introduction.

4) “In this regard (…) NMR spectroscopy (…)” (line 114). “Marvelously, (…) nuclear magnetic resonance (NMR) investigations” (lines 219-220). The abbreviation should be defined the first time that the full-name appears in the text. This comment should be taken into account for the rest of the main manuscript sections.

Response: thanks so much and I have revised them all.

5) “Protein solubility is essential (…) protein aggregation and amyloidogenesis (…) like

Alzheimer’s disease (AD), Parkinsons’s disease (PD) (…) and frontotemporal dementia (FTD)” (lines 169-175). Here, even if I agree with the information provided by the author it should be also remarkable to discuss how other key factors as the presence of certain divalent cations [1] or the ionic strength [2] can also lead to the formation of amyloidogenic fibrils. This will strengthen the multiparametric factors that can trigger the onset of neurodegenerative malignancies and the importance to study them.

Response: thanks so much for the critical suggestion. I have added a short discussion together with the two references.

[1] Carapeto, A.P.; et al. Morphological and Biophysical Study of S100A9 Protein Fibrils by

Atomic Force Microscopy Imaging and Nanomechanical Analysis. Biomolecules 2024, 14, 1091. https://doi.org/10.3390/biom14091091

[2] Ziaunys, M.; et al. Polymorphism of Alpha-Synuclein Amyloid Fibrils Depends on Ionic

Strength and Protein Concentration. Int. J. Mol. Sci. 2021, 22, 12382.

https://doi.org/10.3390/ijms222212382

6) “3. Transformation of the folded cytosolic proteins into membrane-interacting proteins” (lines 328-393). A schematic representation will benefit the potential readers to have a more complete overview of the crosstalk among these two protein states.

Response: thanks so much for the critical suggestion. I have modified Figure 2B as well as developed a paragraph to provide a more complete overview.

7) “10. Summary and challenges (lines 1207-1372). This section perfectly remarks the most

relevant outcomes found by the author in this field and the promising future perspectives and some limitations to be overcome. It should be desirable to add a brief statement to discuss about the potential future action lines to pursue the topic covered in this research.

Response: thanks so much for the critical suggestion. I have added two paragraphs into the last section to address this point.
